# Robust Reinforcement Learning in Finance: Modeling Market Impact with Elliptic Uncertainty Sets

**Shaocong Ma**
Department of Computer Science
University of Maryland
College Park, MD 20742, USA
scma0908@umd.edu

**Heng Huang***
Department of Computer Science
University of Maryland
College Park, MD 20742, USA
heng@umd.edu

## Abstract

In financial applications, reinforcement learning (RL) agents are commonly trained on historical data, where their actions do not influence prices. However, during deployment, these agents trade in live markets where their own transactions can shift asset prices, a phenomenon known as market impact. This mismatch between training and deployment environments can significantly degrade performance. Traditional robust RL approaches address this model misspecification by optimizing the worst-case performance over a set of uncertainties, but typically rely on symmetric structures that fail to capture the directional nature of market impact. To address this issue, we develop a novel class of elliptic uncertainty sets. We establish both implicit and explicit closed-form solutions for the worst-case uncertainty under these sets, enabling efficient and tractable robust policy evaluation. Experiments on single-asset and multi-asset trading tasks demonstrate that our method achieves superior Sharpe ratio and remains robust under increasing trade volumes, offering a more faithful and scalable approach to RL in financial markets.

## 1 Introduction

Reinforcement learning (RL) has emerged as a promising decision-making framework for quantitative trading strategies [1, 2], including portfolio optimization [3–13], automatic trading [3, 14–19], market making [20–22], and option hedging [23, 24]. RL's appeal in finance lies in its ability to learn adaptive strategies directly from data, without strong market assumptions [1]. This makes it well-suited for capturing complex market dynamics and aligning with the sequential nature of financial decision-making.

One of the primary challenges in training a robust and consistently profitable RL agent lies in handling the *market impact* [25–27]; that is, the influence of the agent's own trades on asset prices in the deployed environment. For example, when a trader buys or sells a large volume of an asset, it can temporarily drive the price up or down, respectively (as illustrated in Figure 1). Typically, RL agents are trained on historical market data where market impact is absent. However, during deployment, the environment shifts from a passive historical setting to the real market, where the agent's actions actively affect prices. This discrepancy between training and deployment environments undermines the optimality and robustness of the learned policy, and leads us to the central question in this paper:

> **Q**: *Can we train RL agents on historical data while robustly accounting for market impact during deployment?*

---
*This work was partially supported by NSF IIS 2347592, 2348169, DBI 2405416, CCF 2348306, CNS 2347617, RISE 2536663.

39th Conference on Neural Information Processing Systems (NeurIPS 2025).

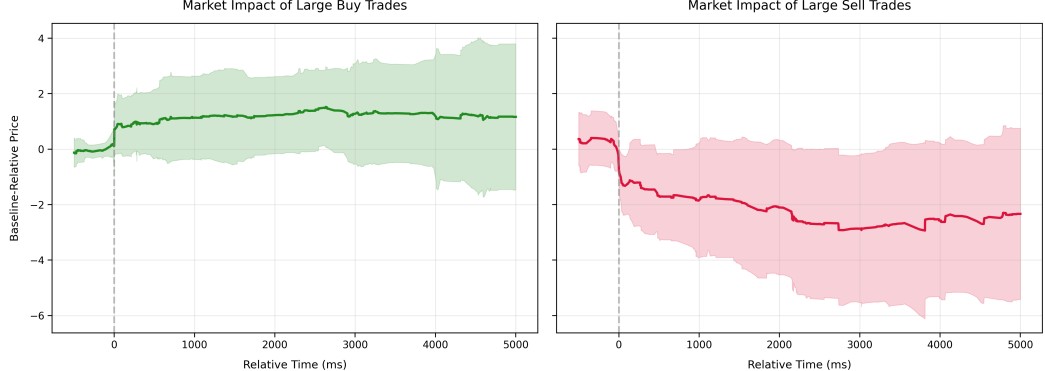

Figure 1: Market impact illustration using AMZN stock on June 21, 2012, based on 5-level LOBSTER data [47]. The left panel shows the price response to executing a buy order of 100 shares at the time $t = 0$ ms, which consumes ask-side liquidity within 1000 ms and induces an immediate upward shift in price. The right panel shows the analogous impact of a sell order. This plot indicates that the transition dynamics induced by trading are not symmetrically distributed around the nominal kernel.

To address this central question, we adopt the framework of robust RL [28–46], which is designed to handle *model misspecification*; that is, the mismatch between the training and deployment environments. The robust RL framework explicitly acknowledges that the real market environment may differ from the simulated training environment, and it aims to learn policies that are resilient under a range of perturbations.

However, existing robust RL approaches face a key limitations in financial applications; that is, traditional uncertainty sets are typically symmetric, which fail to capture the directional nature of market impact. Addressing this challenge motivates our threefold **contributions**:

(1) We propose a novel class of uncertainty sets, *elliptic uncertainty sets* (Definition 3.2), which generalize traditional $\ell_p$-norm balls to the ellipse-like structure. These sets better capture the empirically observed directional nature of market impact as illustrated in Figure 1.

(2) On the theoretical side, we derive closed-form solutions for solving the worst-case transition kernel under the proposed uncertainty sets (Theorem 3.4). Furthermore, under certain conditions, we present the explicit solutions (Theorem 3.5). This development significantly broadens the scope of tractable robust RL problems beyond symmetric (ball-shaped) uncertainty sets, enabling more faithful representation of market impact.

(3) We empirically evaluate our approach on real-world financial data using trade-level market impact simulations in Section 4. Experimental results demonstrate that our method consistently outperforms the standard single-asset intra-day trading strategy and existing RL baselines in terms of Sharpe ratio. Moreover, we validate the robustness of our method under increasing strategy volume, confirming its effectiveness in high-volume regimes.

## 1.1 Related Work

**Existing Approaches to Handle Market Impact**    The most common approach for handling market impact is to simulate the electronic market more accurately. By applying high-fidelity market simulators, often built upon limit order book (LOB) dynamics [25, 26, 48–50], trade-level data [51, 52], or large-scale agent-based simulators [53–56], it captures more detailed market microstructure and reduces the gap between the simulated and the real trading environments. Prominent approaches to incorporating market impact into backtesting include agent-based simulation frameworks [53–56], data-driven LOB reconstruction models [57, 58, 50], and hybrid systems that integrate historical data replay with synthetic order flow generation [59, 60]. However, access to high-quality market data is often limited, and simulating market environments with agent-based systems remains prohibitively expensive. Therefore, a practical alternative is to train the agent directly on historical data without market impact while still encouraging it to account for potential worst-case scenarios.

**Robust RL in Finance**   Robust RL, with its intrinsic ability to handle model misspecification, provides a natural framework for incorporating market impact considerations during training. Jaimungal et al. [3] propose a robust reinforcement learning framework based on rank-dependent utility to address uncertainty in financial decision-making, demonstrating the effectiveness of robust RL in portfolio allocation, benchmark strategy optimization, and statistical arbitrage. Shi et al. [5] formulate portfolio optimization as a robust RL problem to enhance resilience. We et al. [24] extend robust risk-aware RL to manage the risks associated with path-dependent financial derivatives, showcasing its effectiveness in complex hedging scenarios. However, existing work primarily focuses on fully symmetric uncertainty sets, which fail to capture the directional characteristics of financial markets. Addressing this limitation is the central focus of our paper.

**Modeling the Uncertainty Set in Robust RL**   The uncertainty set captures the discrepancy between training and deployment environments. However, robust RL becomes computationally intractable when the uncertainty set is highly irregular [28, 44, 39, 40]. To mitigate this issue, it is common to impose structural assumptions that enable tractable solutions. We highlight several representative structures, with further details deferred to Appendix A.1. The $R$-contamination model [31] defines an uncertainty set as a sphere of radius $R$ centered around the nominal transition kernel, admitting analytical solutions for robust policy evaluation. The $\ell_p$-norm uncertainty sets are also widely studied due to their closed-form solutions [39, 40]. The integral probability metric (IPM) and double-sampling uncertainty sets have been shown to allow efficient computation [43]. Other works include uncertainty sets based on the Wasserstein metric and $f$-divergence, which have also received considerable attention [61–63].

## 2   Preliminaries: Robust Reinforcement Learning

We focus on the discounted infinite-horizon Markov Decision Processes (MDPs) [64], formally defined as a five-tuple $(\mathcal{S}, \mathcal{A}, \mathbb{P}, r, \gamma)$, where $\mathcal{S}$ and $\mathcal{A}$ denote the state and action spaces[2], respectively. The transition kernel $\mathbb{P}(s' \mid s, a)$ specifies the probability of transitioning to state $s'$ from state $s$ after taking action $a$. The reward function $r : \mathcal{S} \to [0, 1]$ assigns a bounded reward to each state, and the discount factor $\gamma \in (0, 1)$ models the agent's preference for immediate rewards over future ones.

Instead of assuming a fixed transition kernel $\mathbb{P}$, we account for the effects of *model misspecification*. Let $\mathbb{P}_0$ denote the nominal transition kernel, and define the $(s, a)$-*uncertainty set* $\mathcal{U}_{s,a} \subset \mathbb{R}^{|\mathcal{S}|}$ at the point $(s, a) \in \mathcal{S} \times \mathcal{A}$ as

$$((s,a)\text{-Uncertainty Set})   \quad \mathcal{U}_{s,a} := \{u_{s,a} \in \mathbb{R}^{|\mathcal{S}|} \mid u_{s,a} \in C_{s,a},\ u_{s,a}^\top \mathbf{1}_{|\mathcal{S}|} = 0\},$$

where $C_{s,a} \subset \mathbb{R}^{|\mathcal{S}|}$ is a convex set, $\mathbf{1}_{|\mathcal{S}|}$ is an all-one vector with the dimension $|\mathcal{S}|$ (for convenience, we will usually omit the subscript $|\mathcal{S}|$). The convex set is commonly chosen as a ball-shaped set (e.g. $C_{s,a} = B_p(\beta_{s,a}) := \{u_{s,a} \in \mathbb{R}^{|\mathcal{S}|} \mid \|u_{s,a}\|_p \leqslant \beta_{s,a}\}$ for the $\ell_p$-norm uncertainty set), where $\beta_{s,a} \geqslant 0$ is a radius parameter that quantifies the allowable deviation. The zero-sum constraint $u_{s,a}^\top \mathbf{1}_{|\mathcal{S}|} = 0$ ensures the perturbed transition $\mathbb{P}(\cdot \mid s, a) + u_{s,a}$ remains a valid probability distribution. The *uncertainty set $\mathcal{U}$* and the *robust transition model* are then defined as

$$\text{(Uncertainty Set)} \quad \mathcal{U} := \bigtimes_{(s,a)\in\mathcal{S}\times\mathcal{A}} \mathcal{U}_{s,a}, \tag{1}$$

$$\text{(Robust Transition Model)} \quad \mathcal{P}_{\mathcal{U}} := \left\{ \mathbb{P}_u := \mathbb{P}_0 + u \,\middle|\, u \in \mathcal{U} \right\}, \tag{2}$$

respectively. Throughout this paper, we assume the parameter of uncertainty set is chosen appropriately such that all elements in the robust transition model is well-defined [39, 40]. Given these notations in place, we define the value function of the policy $\pi$ with the uncertainty $u$ as the value

---

[2]For theoretical analysis, we restrict our attention to MDPs with finite state and action spaces. This assumption avoids the technical complications arising from continuous or hybrid spaces, which, although explored in some prior work [65–68], remain analytically open without imposing additional assumptions, especially for robust RL problems. Nevertheless, our empirical results extend to continuous settings, demonstrating the practical applicability of our approach beyond the theoretical scope.

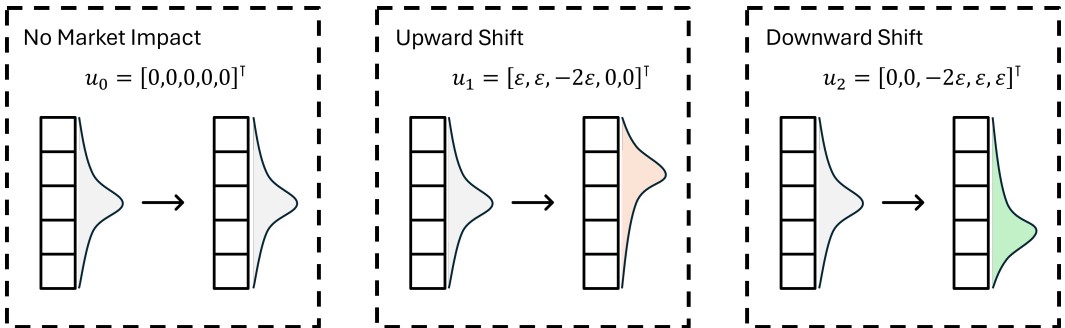

Figure 2: Illustration of the transition kernel in a simplified robust RL setting using an $\ell_\infty$-norm uncertainty set. Only one of the upward (buy) or downward (sell) shift is plausible in a given scenario; however, the ball-shaped uncertainty set must include both shifts due to its symmetric structure, which motivates us to propose an uncertainty set with non-symmetric structures.

function with the transition probability $\mathbb{P}_u \in \mathcal{P}$:

$$V_u^\pi(s) := \mathbb{E}\Big[ \sum_{t=0}^\infty \gamma^t r(s_t, a_t) \mid s_0 = s, \mathbb{P}_u, \pi \Big],$$

The robust value function is the worst-case value function over all uncertainties; that is $V^\pi(s) := \min_{u \in \mathcal{U}} V_u^\pi(s)$. Similarly, we can also define the robust Q-function and the robust advantage function as $Q^\pi(s,a) := \min_u \mathbb{E}\big[ \sum_{t=0}^\infty \gamma^t r(s_t, a_t) \mid s_0 = s, a_0 = a, \mathbb{P}_u, \pi \big]$ and $A^\pi(s,a) := Q^\pi(s,a) - V^\pi(s)$, respectively.

The goal of robust RL is to learn a parameterized policy $\pi_\theta$ that maximizes the worst-case value function $V^{\pi_\theta}(s_0)$, where $s_0$ denotes the initial state. A standard approach applies the robust policy gradient formula [28]:

$$\nabla_\theta V^\pi(s) = \mathbb{E}_{s \sim d^{\pi_\theta}, a \sim \pi_\theta} \left[ Q^{\pi_\theta}(s,a) \nabla_\theta \log \pi_\theta(a|s) \right],$$

where $d^{\pi_\theta}$ is the stationary distribution induced by $\pi_\theta$, and $Q^{\pi_\theta}$ denotes the robust Q-function. This formulation reduces robust RL to a gradient-based optimization problem, shifting the main challenge of robust RL to accurately evaluate the robust value function, which is our focus in Section 3.3.

## 3  Modeling Market Impact with Elliptic Uncertainty Sets

### 3.1  Limitations of Symmetric Uncertainty Sets

Robust MDPs offer an ideal framework to handle model misspecification by allowing the transition kernel to deviate within a prescribed uncertainty set. However, most existing formulations adopt symmetric structures, typically the ball defined by a specific norm, that treat all directions of perturbation equally. While mathematically convenient, these symmetric structures often fail to reflect the directional nature of real-world uncertainties.

Symmetry in this context typically refers to invariance under the signed permutation group (see Appendix A.2 for details). A canonical example is the $\ell_p$-norm ball:

$$B_p(\beta) := \{u \in \mathbb{R}^d \mid \|u\|_p \leq \beta\},$$

which satisfies the property that for any $u \in B_p(\beta)$, all signed permutations of $u$ are also contained in the set. It enforces an implicit assumption of isotropic uncertainty, equally plausible in all directions, which often includes unrealistic perturbations. We demonstrate it in the following example:

**Example 3.1** (Symmetric Sets Fail to Capture Directional Uncertainty)**.** In financial markets, large buy or sell orders induce directional shifts in asset prices due to liquidity consumption (see Figure 1). In Figure 2, we consider the following classical $\ell_\infty$-norm $(s,a)$-uncertainty set:

$$\mathcal{U}_{s,a} := \left\{ u \in \mathbb{R}^d \mid \|u\|_\infty \leq 2\varepsilon, \ u^\top \mathbf{1} = 0 \right\}.$$

This set includes, for example, the perturbation vectors

$$u_1 = [\varepsilon, \varepsilon, -2\varepsilon, 0, 0]^\top, \quad \text{and} \quad u_2 = [0, 0, -2\varepsilon, \varepsilon, \varepsilon]^\top.$$

Both $u_1$ and $u_2$ satisfy the norm and mean constraints, and since they are signed permutations of each other, they must either both belong to $\mathcal{U}_{s,a}$ or be excluded together. However, this symmetry fails to reflect market realities: under a buy action, $u_1$ represents a plausible upward shift due to liquidity-driven market impact, while $u_2$, corresponding to a downward shift, is implausible. Thus, the symmetric structure forces inclusion of perturbations that contradict the directional market impact, potentially leading to overly conservative or unrealistic robust policies.

This observation underscores the importance of developing a robust RL framework that can capture the directional nature of environment shifts observed in financial markets. To this end, we introduce a novel class of *elliptic uncertainty sets*, which generalize traditional norm-bounded sets by allowing non-symmetric perturbations, while retaining closed-form tractability under certain conditions.

**On the Conservativeness of Robust Policies**  Although our proposed elliptic uncertainty sets better capture the directional nature of market impact, this alone does not immediately clarify why they improve robustness. To illustrate the underlying intuition, we present a simple example:

- The robust value function $V^\pi := \min_{u \in \mathcal{U}} V_u^\pi$ models the worst-case discounted future return. If the uncertainty set $\mathcal{U}$ is larger, it is more conservative, as it yields smaller value.

- In the ideal case, the uncertainty set consists of a single element $u_0$ that exactly characterizes the MDP induced by the true market impact. Training an RL agent on the robust MDP with $\mathcal{U} = u_0$ is then equivalent to training directly on the real LOB data.

- In the less ideal case where $\mathcal{U}$ includes additional elements, *e.g.* $\mathcal{U} = \{u_0, u_1\}$. The robust value function may still achieve its minimum at $u_0$ ($u_0 = \arg\min_u \min_{u \in \mathcal{U}} V_u^\pi$). In this scenario, the robust formulation reduces to the ideal case. Otherwise, if the minimum is attained at some $u \neq u_0$, the robust value becomes strictly smaller than $V_{u_0}^\pi$, making the policy more conservative.

Unfortunately, neither our approach nor standard robust RL methods can theoretically rule out this latter "overly conservative" scenario. Our guiding intuition, however, is that smaller uncertainty sets are less likely to contain such overly conservative elements. By trimming down the traditional uncertainty sets, our method reduces the risk of unnecessary conservatism, though it does not eliminate it entirely.

## 3.2 The Elliptic Uncertainty Sets

The *elliptic uncertainty sets* generalize the classical $\ell_p$-norm uncertainty set by incorporating directional non-symmetry. Formally, we have the following definition:

**Definition 3.2** (Elliptic $(s, a)$-Uncertainty Set). For each state-action pair $(s, a) \in \mathcal{S} \times \mathcal{A}$, the **elliptic $(s, a)$-uncertainty set** is defined as

$$\mathcal{U}_{s,a} := \left\{ u \in \mathbb{R}^{|\mathcal{S}|} \;\middle|\; \sum_{n=1}^{N} \|u - u_n^{s,a}\| \leq \beta_{s,a}, \; u^\top \mathbf{1}_{|\mathcal{S}|} = 0 \right\}, \tag{3}$$

where $\{u_n^{s,a}\}_{n=1}^N \in \mathbb{R}^{|\mathcal{S}|}$ are called the *foci* of the ellipse, $\beta_{s,a} \geq 0$ is the *uncertainty size*, and $\|\cdot\| : \mathbb{R}^d \to \mathbb{R}$ is an arbitrary norm. Particularly, when $\|\cdot\|$ is taken as the $\ell_p$-norm ($p \in [1, +\infty]$), $\mathcal{U}_{s,a}$ is called the $\ell_p$-**ellipse uncertainty set**.

Here the constraint $u^\top \mathbf{1}_{|\mathcal{S}|} = 0$ ensures the perturbed transition still defines a valid probability distribution. For convenience, we will omit the superscript in $u_n^{s,a}$ and the subscript in $\beta_{s,a}$ when the context is clear. Importantly, we note that it is unavoidable that the element in $\mathcal{U}_{s,a}$ may not be a valid probability distribution when some entries of $u + \mathbb{P}_0(\cdot|s, a)$ are negative. To avoid this scenario, we include the following regular condition, which is also presented in the $\ell_p$-norm uncertainty set literature [39, 40]:

**Assumption 3.3.** *For the given MDP $(\mathcal{S}, \mathcal{A}, \mathbb{P}_0, r, \gamma)$, for any set of foci $\{u_n^{s,a}\}_{n=1}^N$, there exists a constant $\overline{\beta} > 0$ such that for all $\beta_{s,a} < \overline{\beta}$, Eq. (3) induces valid probability transitions; that is, all entries of $u + \mathbb{P}_0(\cdot|s, a)$ are non-negative.*

**Connection to the Classical Ellipse**  Our definition draws directly from the geometric characterization of an ellipse: the set of points for which the sum of distances to multiple foci is bounded by a constant $\beta_{s,a}$. We show that it recovers classical structures as special cases with the following two concrete examples:

(1) When $N = 1$ and $u_1 = 0$, the set Eq. (3) reduces to the $\ell_p$-norm ball
$$B_p(\beta) := \{u \mid \|u\|_p \leqslant \beta\},$$
which aligns with the standard uncertainty set used in robust RL (e.g., [39, 40, 28]).

(2) When $N = 2$ and $p = 2$, Eq. (3) becomes $\|u - u_1\|_2 + \|u - u_2\|_2 \leqslant \beta$. Defining the midpoint $\bar{u} := \frac{u_1 + u_2}{2}$, there exists a matrix $A$ such that this constraint is equivalent to a classical quadratic form of an ellipse:
$$(u - \bar{u})^\top A(u - \bar{u}) \leqslant 1,$$
where the detailed derivation is given in Lemma B.10.

These examples demonstrate that our formulation significantly generalizes classical ellipses by allowing arbitrary norms and accommodating multiple foci, thereby enabling more flexible modeling of non-symmetric perturbations. However, such generality often comes at the cost of increased complexity. To address this concern, in the next section, we show that, under certain parameter choices, our proposed uncertainty set remains as trackable as traditional $\ell_p$-norm uncertainty sets.

**Limitations & Potential Directions**  As our approach is mainly adopted from the $\ell_p$-norm uncertainty set, it does not guarantee that the resulting distributions are absolutely continuous with respect to the nominal transition distribution even with Assumption 3.3. One potential solution is incorporating $f$-divergence or distributionally robust RL techniques [69–74].

### 3.3  Solving the Worst-Case Uncertainty

Solving the worst-case uncertainty $u^* := \arg\min V_u^\pi(s_0)$ plays a crucial role in efficient robust policy evaluation. The $\ell_p$-norm uncertainty set [39, 40] and the $R$-contamination model [31] are popular as the solution $u^*$ is closed-form. When the uncertainty set is complicated, many existing robust RL methods require an external optimization loop to determine the worst-case transition probability, which can be impractical in real-world scenarios.

To address this issue, we derive an implicit solution (Theorem 3.4) for the worst-case uncertainty that avoids additional interaction with the environment. Under certain conditions, we further provide an explicit closed-form solution (Theorem 3.5), enabling direct computation without the need for external solvers.

We start with recapping some backgrounds in robust TD learning. Let $\mathcal{V}$ denote all functions mapping from the state space $\mathcal{S}$ to the Euclidean space $\mathbb{R}$. Given that $\mathcal{U}$ is an arbitrary uncertainty set, the robust Bellman operator associated with the policy $\pi$, $\mathscr{T}_{\mathcal{U}}^\pi : \mathcal{V} \to \mathcal{V}$, is defined as

$$\mathscr{T}_{\mathcal{U}}^\pi v(s) = \sum_{a \in \mathcal{A}} \pi(a|s) \left[ r(s,a) + \gamma \min_{u \in \mathcal{U}_{s,a}} u^\top v + \gamma \sum_{s' \in \mathcal{S}} \mathbb{P}_0(s'|s,a)v(s') \right].$$

As shown by [75, 28], when $\gamma \in (0, 1)$, the robust Bellman operator is a contraction operator, which admits the unique fixed point as the robust value function $V^\pi \in \mathcal{V}$. When $\mathcal{U}$ is given by the $\ell_p$-ellipse uncertainty set (Eq. (3)), then

$$\mathscr{T}_{\mathcal{U}}^\pi v(s) = \sum_{a \in \mathcal{A}} \pi(a|s) \left[ r(s,a) + \gamma \min_{\substack{\sum_n \|u - u_n\| \leqslant \beta \\ u^\top \mathbf{1} = 0}} u^\top v + \gamma \sum_{s' \in \mathcal{S}} \mathbb{P}_0(s'|s,a)v(s') \right].$$

It turns out that if we can solve the optimization problem

$$\min_{\substack{\sum_n \|u - u_n\| \leqslant \beta_{s,a} \\ u^\top \mathbf{1} = 0}} u^\top v, \tag{4}$$

the robust Bellman operator is just the standard Bellman operator (over the nominal transition probability) with a solved shift. As the result, we can simply apply the standard TD-learning with adding this correction term to solve the desired robust value function. In the following theorem, we present a general recipe of solving this optimization problem.

**Theorem 3.4** (Implicit). *Let $d := |\mathcal{S}|$ be the cardinal of state space. Suppose that $\{u_n\}_{n=1}^N \subset \mathbb{R}^d$ for each $(s,a) \in \mathcal{S} \times \mathcal{A}$, the uncertainty size $\beta \geqslant 0$, and $v \in \mathbb{R}^d$. Then there exists $\lambda^*$ and $\mu^*$ such that the optimization problem defined by Eq. (4) is solved by*

$$u^* = \arg\min_u \Big[(v + \mu^* \mathbf{1})^\top u + \lambda^* \sum_{n=1}^N \|u - u_n\|_p\Big].$$

*Remark.* We say a solution of the optimization problem is "implicit" if it can be represented as an equation in the form $g(u, v, \beta, \{u_n\}) = 0$. In this theorem, as the right-hand side

$$G(u, v, \beta, \{u_n\}) := (v + \mu^* \mathbf{1})^\top u + \lambda^* \sum_{n=1}^N \|u - u_n\|_p$$

is proper, convex, and coercive; we can surely re-write it as the sub-gradient form by letting

$$g(u, v, \beta, \{u_n\}) \in \partial_u G(u, v, \beta, \{u_n\}).$$

Then we obtain the implicit representation $g(u, v, \beta, \{u_n\}) = 0$. Moreover, we derive the formula of $\lambda^*$ and $\mu^*$ beyond the existence; the full result is presented in Theorem B.12 with more details.

The implicit solution has already shown significant advances compared to some of existing robust RL methods which typically require to solve the worst-case transition probability using additional state-action sample generated from the agent-environment iteration. However, it is still (slightly) impractical to solve additional convex optimization problems in every iteration. Fortunately, under certain conditions, the solution can be "explicit"; that is, we can write the optimal solution in the form of $u^* = f(v, \beta, \{u_n\})$.

**Theorem 3.5** (Explicit). *Let $d := |\mathcal{S}|$ be the cardinal of state space. Suppose that $\{u_n\}_{n=1}^N \subset \mathbb{R}^d$ for each $(s,a) \in \mathcal{S} \times \mathcal{A}$, the uncertainty size $\beta \geqslant 0$, and $v \in \mathbb{R}^d$. The optimization problem defined by Eq. (4) is explicitly solved in the following cases:*

*(a) Let $N = 1$ and $\frac{1}{p} + \frac{1}{q} = 1$. Then the minimizer of Eq. (4) is given by*

$$u^* = u_1 + \beta \frac{\text{sign}(v + \mu^* \mathbf{1}) \odot |v + \mu^* \mathbf{1}|^{q-1}}{\|v + \mu^* \mathbf{1}\|_q^{q-1}}.$$

*Here $\text{sign}$ and $|\cdot|$ are coordinate-wise sign function and absolute value, respectively; $\odot$ is the coordinate-wise product; and $\mu^* := \arg\min_{\mu \in \mathbb{R}} \|v + \mu \mathbf{1}\|_q$.*

*(b) Let $p = 1$ and $N = 2$. Suppose that $\beta > \|u_2 - u_1\|_1$. Then the minimizer of Eq. (4) is given by*

$$u^* = \frac{u_1 + u_2}{2} - \frac{\beta - \|u_1 - u_2\|_1}{2} \Big[\text{sign}(v + \mu^* \mathbf{1}) \odot \mathbb{I}_{\{|v + \mu^* \mathbf{1}| = 2\lambda^*\}}\Big].$$

*Here $\text{sign}$, $|\cdot|$, and $\mathbb{I}$ are coordinate-wise sign function, absolute value, and indicator function, respectively; $\odot$ is the coordinate-wise product;*

$$\mu^* := -\frac{\max_j v_j + \min_j v_j}{2}, \quad \text{and} \quad \lambda^* := -\frac{\max_j v_j - \min_j v_j}{4}.$$

*(c) Let $p = 2$ and $N = 2$. Suppose that $\beta > \|u_2 - u_1\|_2$. Then the minimizer of Eq. (4) is given by*

$$u^* = \frac{u_1 + u_2}{2} - \frac{\sqrt{\beta^2 - \|u_2 - u_1\|_2^2}}{2} \frac{1}{\underline{\lambda}^*} \Omega^{-1}(v + \mu^* \mathbf{1}).$$

*Here $\Omega := I - \frac{1}{\beta^2}(u_2 - u_1)(u_2 - u_1)^\top$;*

$$\underline{\lambda}^* = \sqrt{(v + \mu^* \mathbf{1})^\top \Omega^{-1}(v + \mu^* \mathbf{1})}, \quad \text{and} \quad \mu^* = -\frac{v^\top \Omega^{-1} \mathbf{1}}{\mathbf{1}^\top \Omega^{-1} \mathbf{1}}.$$

*Remark.* This result presents a clean form of the worst-case uncertainty $u^*$ under certain conditions. Unlike the implicit case where it takes an additional root-finding algorithm to solve an approximated $u^*$, in the explicit case, if the current value function $v \in \mathcal{V}$ is given and the parameters of the uncertainty set ($u_i$ and $\beta$) have been determined, the uncertainty $u^*$ can be explicitly solved. The full proof is presented in Theorem B.13.

### 3.4 Robust TD Learning Algorithm

Given Theorem 3.4 and Theorem 3.5 in place, we immediately obtain the robust TD learning algorithm for robust policy evaluation. Given the current value function $v$, we can calculate $u^*$ using the implicit and the explicit formula; assume the current state-action pair is given as $(s, a, s')$, then the updated value function is given by

$$v'(s) = v(s) + \eta \left( r(s, a) + \gamma \sum_{s'} \left( \mathbb{P}_0(s'|s, a) + u^*(s') \right) \left[ v(s') + \gamma v(s') \right] - v(s) \right). \quad (5)$$

We can further simplify this update rule by using an unbiased estimator of that:

$$v'(s) = v(s) + \eta \gamma v^\top u^* + \eta \left( r(s, a) + \gamma v(s') - v(s) \right), \quad (6)$$

where $s' \sim \mathbb{P}_0(\cdot|s, a)$. The formulation leads us to Algorithm 1.

---

**Algorithm 1:** Robust Policy Evaluation

**Input:** The target policy $\pi$, the foci $\{p_i\}_{i=1}^N \subset \mathbb{R}^d$, and $\{\beta_{s,a}\}$ the uncertainty size

Sample the initial state $s_0$ from the initial distribution;
Initialize the value function $v_0$;
**for** $t = 0, 1, 2, \ldots, T - 1$ **do**
    Sample the action $a_t \sim \pi(\cdot|s_t)$; Transition from $s_t$ to $s_{t+1} \sim \mathbb{P}_0(\cdot|s_t, a_t)$;
    Calculate $u^*$ using $\{p_i\}_{i=1}^N$, $\{\beta_{s,a}\}$, and $v_t$;
    Robust TD-learning: $v'(s) = v(s) + \eta \gamma v^\top u^* + \eta \left( r(s, a) + \gamma v(s') - v(s) \right)$;
**end**

**Output:** The final value function $v_T$

---

*Remark.* The convergence of this algorithm, as well as its corresponding Actor-Critic-style policy gradient algorithm, follows directly by applying the standard proof routine from the robust RL literature (e.g. [43]). For completeness, we include the convergence result and full proof in the supplementary material.

## 4 Experiments

To validate our theoretical findings and demonstrate the practical effectiveness of robust RL framework in the environment with the market impact, we conduct experiments on two different tasks that are closely tied to market impact: (1) minute-level single-asset strategy, and (2) large-volume portfolio rebalancing. In minute-level trading, even small trade sizes can noticeably move prices, leading to slippage. Similarly, large-scale portfolio rebalancing, often performed by large financial institutions, can significantly affect asset prices due to the large order volumes involved.

### 4.1 Performance Comparison on Single-Asset Intra-Day Trading

We start with the single-asset minute-level trading. The non-RL baseline is chosen as the momentum strategy [76], which is designed based on the empirical observation where assets that have performed well in the recent past are more likely to continue performing well in the near future.

**Training and Evaluation of RL Agents** We implement a Gym-like RL environment [77, 78] constructed on historical data, with full environment details provided in Appendix C.3. All RL agents are trained on one year of earlier historical data (from May 9th, 2021 to May 9th, 2022 as the nominal environment) without accounting for market impact. Their performances are then evaluated over the period from June 9 to December 9, 2022, with the market impact included. To simulate market impact, we reconstruct LOB dynamics using a short period of real trading orders and determine the execution price via the volume-weighted average price (VWAP). A simple example illustrating this estimation process is shown in Table 2, Appendix C.

**Results** As shown in Table 1, our proposed method consistently outperforms the momentum strategy, the non-robust RL, and the symmetric robust RL baselines (based on $\ell_p$-norm balls) in terms of the

Table 1: Performance comparison of different RL agents on selected assets under the simulated market impact from June 9 to December 9, 2022. Robust RL with $\ell_p$-ellipse uncertainty set consistently achieves the highest Sharpe ratio.

| Asset | Method | Final Value ($) | Annualized Return (%) | Sharpe Ratio | Max Drawdown (%) |
|-------|--------|-----------------|------------------------|--------------|-------------------|
| META | Momentum | 96 334 | −7.3% | −0.95 | −4.2% |
| | Non-Robust RL | 120 347 | 44.8% | 1.74 | −11.3% |
| | Robust RL ($\ell_p$-Ball) | 97 103 | −5.7% | −0.28 | −13.4% |
| | Robust RL ($\ell_p$-Ellipse) | 138 011 | 90.8% | 2.48 | −9.9% |
| MSFT | Momentum | 105 163 | 10.6% | 1.10 | −5.3% |
| | Non-Robust RL | 87 440 | −23.5% | −1.75 | −16.9% |
| | Robust RL ($\ell_p$-Ball) | 92 159 | −15.1% | −0.82 | −11.2% |
| | Robust RL ($\ell_p$-Ellipse) | 111 485 | 24.4% | 1.20 | −10.1% |
| SPY | Momentum | 107 333 | 15.2% | 1.69 | −3.2% |
| | Non-Robust RL | 91 947 | −15.5% | −1.64 | −11.9% |
| | Robust RL ($\ell_p$-Ball) | 100 560 | 1.1% | 0.17 | −6.4% |
| | Robust RL ($\ell_p$-Ellipse) | 109 272 | 19.4% | 1.60 | −5.8% |

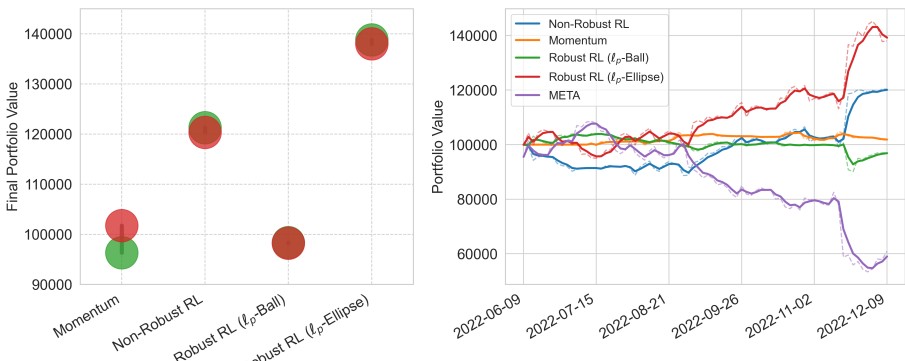

Figure 3: Performance comparison of trading strategies on the META stock from June 9 to December 9, 2022, under simulated market impact. The left panel compares the final portfolio values with (in red) and without market impact (in green), illustrating the robustness of each method to execution-related slippage. The right panel shows the cumulative returns over the evaluation period, tracking the performance of the four strategies in Table 1, alongside the baseline performance of the META stock.

risk-adjusted return (Sharpe Ratio). These experiments validate the following key understandings: *(i)* While robust RL with symmetric uncertainty sets significantly mitigates the effects of market impact (as illustrated in the left panel of Figure 3), it often produces overly conservative strategies that compromise profitability by taking implausible perturbations into consideration; *(ii)* the non-robust RL usually suffers greater risk exposure, resulting in the highest Max Drawdown among all methods; *(iii)* in contrast, the proposed $\ell_p$-ellipse uncertainty set effectively captures the directional non-symmetry of market impact, allowing the agent to achieve a more favorable trade-off between robustness and return.

## 4.2 Robustness to the Market Impact Scaling in the Trading Volume

In this subsection, we show that a policy trained in a low-volume environment continues to mitigate market impact when transferred to portfolios with significantly larger volumes. We consider a multi-asset portfolio allocation task, modeling the realistic setting where large volumes are traded over short periods to maintain a low-variance portfolio. The same Gym-like RL environment and evaluation period from the previous experiment are used. We evaluate the robustness to the market impact using the relative portfolio gap, the normalized absolute difference in final portfolio value with and without market impact:

$$\text{Relative Port. Gap} = \frac{|\text{Port. Value with MI} - \text{Port. Value without MI}|}{\text{Initial Cash}},$$

where MI represents the market impact. Additional experimental details are provided in Appendix C.

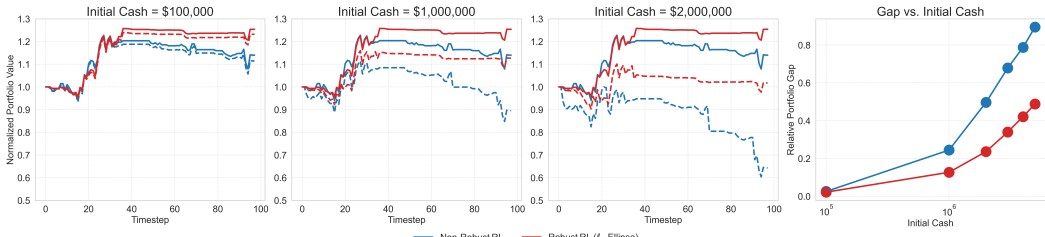

Figure 4: Robustness of RL agents to market impact under increasing trading volumes. The left three panels show normalized portfolio values over time across initial cash levels, with dashed and solid lines indicating performance with and without market impact, respectively. The right panel shows the relative portfolio gap, which increases sharply for the non-robust agent but remains small and stable for the robust RL agent with $\ell_p$-elliptic uncertainty sets.

**Results**   As shown in Figure 4, the robust RL agent with $\ell_p$-ellipse uncertainty set consistently outperforms the non-robust RL method both in return and in mitigating the effects of market impact. While the performance gap is small at low volume, the non-robust agent degrades rapidly as volume increases, suffering from instability and larger drawdowns. In contrast, the robust agent remains stable and profitable even at high volume ($\sim 200$ M), demonstrating strong scalability.

# 5   Conclusion & Broader Impact

This paper focuses on the market impact appearing in quantitative trading, where an agent's actions affect prices. By modeling the training environment as the nominal transition kernel, the proposed novel $\ell_p$-ellipse uncertainty sets better captures the non-symmetric nature of price responses compared to traditional symmetric sets. We established the theoretical tractability of this approach by deriving implicit and explicit closed-form solutions for robust policy evaluation within this framework, enabling efficient robust TD-learning algorithms that account for the market impact during training on the nominal historical environment. Experiments on real historical data demonstrated that our method significantly improves robustness and risk-adjusted returns over non-RL, non-robust RL, and symmetric robust RL baselines. This work broadens the applicability of tractable robust RL and offers a more faithful modeling approach for market impact. The broader impact involves potentially more stable and profitable automated trading strategies.

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

# Appendix

## Table of Contents

## A  Backgrounds

### A.1  Common Uncertainty Sets in the Literature

When evaluating the robust Bellman operator

$$\mathscr{T}_{\mathcal{U}}^{\pi} v(s) = \sum_{a \in \mathcal{A}} \pi(a|s) \left[ r(s,a) + \gamma \min_{\mathbb{P} \in \mathcal{P}_{s,a}} \sum_{s' \in \mathcal{S}} \mathbb{P}(s'|s,a) v(s') \right], \tag{7}$$

the uncertainty set $\mathcal{U} = \{\mathcal{P}_{s,a}\}_{(s,a) \in \mathcal{S} \times \mathcal{A}}$ plays a crucial role. Certain structures in $\mathcal{U}$ enable efficient robust policy evaluation. Below, we summarize several widely adopted constructions.

$f$-**divergence**    The $f$-divergence family [62, 63] generalizes statistical distances between distributions using a convex function $f : (0, \infty) \to \mathbb{R}$ with $f(1) = 0$. For distributions $\mathbb{P}$ and $\mathbb{P}_0$ such that $\mathbb{P} \ll \mathbb{P}_0$, the $f$-divergence is defined as

$$D_f(\mathbb{P} \| \mathbb{P}_0) := \int_{\mathcal{S}} \mathbb{P}_0(s) f\left( \frac{\mathbb{P}(s)}{\mathbb{P}_0(s)} \right) ds.$$

Special cases include the Kullback-Leibler divergence ($f(t) = t \log t$), total variation distance ($f(t) = \frac{1}{2}|t - 1|$), and $\chi^2$-divergence ($f(t) = (t - 1)^2$). In robust RL, the $f$-divergence ball around the nominal transition kernel $\mathbb{P}_0(\cdot|s,a)$ yields the uncertainty set

$$\mathcal{P}_{s,a} = \left\{ \mathbb{P}(\cdot|s,a) \in \Delta^{|\mathcal{S}|} \mid D_f(\mathbb{P}(\cdot|s,a) \| \mathbb{P}_0(\cdot|s,a)) \leqslant \beta_{s,a} \right\}.$$

The inner minimization in Eq. (7) becomes a distributionally robust optimization problem over $\mathbb{P}_0$. As the result, the robust policy evaluation under the KL-divergence often requires to repeatedly solve an additional convex program.

**$R$-contamination Model**    The $R$-contamination model [31] assumes that the true transition kernel lies within a convex mixture of the nominal model $\mathbb{P}_0$ and an arbitrary distribution $\mathbb{P}_1$:

$$\mathcal{P}_{s,a} = \left\{ (1-R)\mathbb{P}_0(\cdot|s,a) + R\mathbb{P}_1(\cdot|s,a) \mid \mathbb{P}_1(\cdot|s,a) \in \Delta^{|\mathcal{S}|} \right\},$$

where $R \in [0,1]$ quantifies the contamination level. This model leads to closed-form solutions for the robust Bellman operator, with the worst-case distribution taking mass at the minimum of the value function $v$. As a result, this setup enables efficient and model-free learning algorithms, including robust variants of Q-learning, TD learning, and policy gradients. It is particularly well-suited for online learning, where $\mathbb{P}_0$ evolves with the observed data.

**$\ell_p$-norm**    These sets constrain the deviation from the nominal model $\mathbb{P}_0(\cdot|s,a)$ using the $\ell_p$-norm:

$$\mathcal{P}_{s,a}^{(p)} = \left\{ \mathbb{P}(\cdot|s,a) \in \Delta^{|\mathcal{S}|} \mid \|\mathbb{P}(\cdot|s,a) - \mathbb{P}_0(\cdot|s,a)\|_p \leqslant \beta_{s,a} \right\}.$$

When $p = 1$, the constraint corresponds to total variation distance, while $p = \infty$ bounds the largest single-coordinate deviation. These sets are commonly used due to their interpretability and explicit analytical solution given in [39, 40]. However, their axis-aligned geometry can lead to overly conservative policies in high dimensions.

**Integral Probability Metric (IPM)**    The IPM measures the discrepancy between distributions through expectations over a function class $\mathcal{F}$:

$$d_{\mathcal{F}}(\mathbb{P}, \mathbb{P}_0) := \sup_{f \in \mathcal{F}} |\mathbb{E}_{\mathbb{P}}[f] - \mathbb{E}_{\mathbb{P}_0}[f]|.$$

The corresponding uncertainty sets are:

$$\mathcal{P}_{s,a} = \left\{ \mathbb{P}(\cdot|s,a) \in \Delta^{|\mathcal{S}|} \mid d_{\mathcal{F}}(\mathbb{P}(\cdot|s,a), \mathbb{P}_0(\cdot|s,a)) \leqslant \beta_{s,a} \right\}.$$

The IPM-based uncertainty sets are particularly useful when the state space is extremely large or continuous, as explicitly solve the minimization problem in Eq. (7) does not requires to access values at all states [43].

**Wasserstein Distance**    The Wasserstein distance [61], grounded in optimal transport theory, accounts for the geometry of the state space. Given a cost function $d : \mathcal{S} \times \mathcal{S} \to \mathbb{R}_+$ and $p \geqslant 1$, the $p$-Wasserstein distance between $\mathbb{P}$ and $\mathbb{P}_0$ is

$$W_p(\mathbb{P}, \mathbb{P}_0) := \left( \inf_{\gamma \in \Gamma(\mathbb{P}, \mathbb{P}_0)} \int_{\mathcal{S} \times \mathcal{S}} d(s,s')^p d\gamma(s,s') \right)^{1/p},$$

where $\Gamma(\mathbb{P}, \mathbb{P}_0)$ denotes the set of joint distributions (couplings) with marginals $\mathbb{P}$ and $\mathbb{P}_0$. The uncertainty set is then

$$\mathcal{P}_{s,a} = \left\{ \mathbb{P}(\cdot|s,a) \in \Delta^{|\mathcal{S}|} \mid W_p(\mathbb{P}(\cdot|s,a), \mathbb{P}_0(\cdot|s,a)) \leqslant \beta_{s,a} \right\}.$$

Despite their strong theoretical properties, solving the inner minimization often requires dual formulations or approximation techniques.

**General Uncertainty Sets**    There are also many techniques developed to handle the situation where the uncertainty set is general. For example, However, [44] proposes a bilevel approach that iteratively solves the worst-case transition kernel to approximate the robust value function. However, as demonstrated in [28, 44, 75], solving robust RL problems in the general case is NP-hard.

## A.2    The Signed Permutation Group

The **signed permutation group** plays a central role in characterizing the symmetry structure of uncertainty sets in our robust RL framework. Informally, this group consists of all matrices in $\mathbb{R}^{|\mathcal{S}| \times |\mathcal{S}|}$ satisfying the following conditions:

1. Each entry is either $0$, $1$, or $-1$.

2. Each row and each column contains exactly one nonzero entry.

In other words, every element of this group is a matrix obtained by permuting the standard basis vectors of $\mathbb{R}^{|\mathcal{S}|}$ and possibly flipping their signs. Each such matrix can be expressed as the product $DP$, where $D$ is a diagonal matrix with diagonal entries in $\{\pm 1\}$, and $P$ is a permutation matrix representing an element of the symmetric group $S_{|\mathcal{S}|}$. This leads to the following formal definition:

**Definition A.1** (Signed Permutation Group). Let $S_{|\mathcal{S}|}$ denote the permutation group over $|\mathcal{S}|$ elements, and let $(\mathbb{Z}_2)^{|\mathcal{S}|}$ be the direct sum of $|\mathcal{S}|$ copies of the cyclic group of order 2. Then the *signed permutation group*, denoted by $\mathrm{Signed}(S_{|\mathcal{S}|})$, is the semidirect product:

$$\mathrm{Signed}(S_{|\mathcal{S}|}) \cong (\mathbb{Z}_2)^{|\mathcal{S}|} \rtimes S_{|\mathcal{S}|},$$

where the action of $S_{|\mathcal{S}|}$ on $(\mathbb{Z}_2)^{|\mathcal{S}|}$ is given by permuting the order.

In this work, we define the "symmetry" of sets as the invariance under the group action induced by the signed permutation group. Specifically, we say a set $B \subset \mathbb{R}^{|\mathcal{S}|}$ is *symmetric under a group action* by $G$ if $g \cdot B \subseteq B$ for all $g \in G$ (or $G \cdot B := \{g \cdot b\}_{g \in G, b \in B} \subseteq B$). This notion of symmetry leads to the following structural property of $\ell_p$-norm balls:

**Proposition A.2.** *Let $\mathcal{B} := \{B_p(\beta)\}_{p \geqslant 1, \beta \geqslant 0}$ be the family of $\ell_p$-norm balls, where $B_p(\beta) := \{u \in \mathbb{R}^d \mid \|u\|_p \leqslant \beta\}$. If there exists a group $G$ such that all elements in $\mathcal{B}$ are symmetric under the group action by $G$, then $G$ must be isomorphic to a subgroup of $\mathrm{Signed}(S_d)$.*

*Proof.* All elements in $\mathcal{B}$ are symmetric under the group action by $G$; that is, for every $p \geqslant 1$ and every $\beta \geqslant 0$,
$$g\big(B_p(\beta)\big) = B_p(\beta) \quad \forall g \in G.$$
Therefore, the act $g : \mathbb{R}^d \to \mathbb{R}^d$ is a norm-preserving bijection; by the Mazur-Ulam theorem, it must be affine. Then as it preserves 0, it must be linear.

In particular, taking $p = 1$ and $\beta = 1$, each $g \in G$ is an (invertible) linear isometry of the 1-norm unit ball
$$B_1(1) = \big\{u \in \mathbb{R}^d : \|u\|_1 \leqslant 1\big\},$$
whose extreme points are exactly
$$\big\{\pm e_1, \pm e_2, \ldots, \pm e_d\big\}.$$
Because a linear automorphism of a polytope must permute its extreme points, for each $i$ and each $g \in G$ there must exist a sign $\varepsilon_i \in \{\pm 1\}$ and an index $\sigma(i) \in \{1, \ldots, d\}$ such that
$$g(e_i) = \varepsilon_i e_{\sigma(i)}.$$
Thus in the standard basis $g$ is represented by a *signed permutation matrix*:
$$g = DP,$$
where $D = \mathrm{diag}(\varepsilon_1, \ldots, \varepsilon_d)$ and $P$ is the permutation matrix corresponding to $\sigma \in S_d$. Hence every $g \in G$ lies in the signed permutation group $\mathrm{Signed}(S_d)$. In other words $G \subseteq \mathrm{Signed}(S_d)$, which equivalently shows $G$ is isomorphic to a subgroup of $\mathrm{Signed}(S_d)$. $\qquad \square$

This result provides a useful insight in designing the $\ell_p$-ellipse set: the signed permutation group is the *largest group* under which all $\ell_p$-norm balls are symmetric. Consequently, to construct a set with less symmetry than the standard $\ell_p$-norm balls (as we aim to do with our $\ell_p$-ellipse sets), it is necessary to enlarge the family $\mathcal{B}$ to include non-ball shapes.

# B Worst-Case Uncertainty under $\ell_p$-Ellipse Uncertainty Sets

## B.1 Supporting Lemmas

**Definition B.1** (Minkowski sum). Given two sets $A, B \subseteq \mathbb{R}^d$, the Minkowski sum $+ : 2^{\mathbb{R}^d} \times 2^{\mathbb{R}^d} \to 2^{\mathbb{R}^d}$ is defined as
$$A + B = \{a + b \mid a \in A, b \in B\}.$$

**Definition B.2** (Fenchel conjugate [79])**.** Let $g : \mathbb{R}^d \to \mathbb{R}$ be a function over $\mathbb{R}^d$. Its Fenchel conjugate is denoted as $g^* : \mathbb{R}^d \to \mathbb{R}$ and is defined as

$$g^*(y) := \sup_{x \in \mathbb{R}^d} \left\{ y^\top x - g(x) \right\}.$$

We include the following famous Hölder's inequality without providing the proof.

**Lemma B.3** (Hölder's inequality)**.** *Let $p, q \in [1, +\infty]$ satisfy $\frac{1}{p} + \frac{1}{q} = 1$. For every $f, g \in \mathbb{R}^d$*

$$f^\top g \leqslant \|f\|_p \|g\|_q.$$

*Moreover, equality holds if and only if*

$$g = 0 \quad or \quad \frac{g}{\|g\|_q} \in J_p(f),$$

*where $J_p(f)$ denotes any $\ell_p$-unit vector that attains the maximum inner product with $f$:*

$$J_p(f) = \arg \max_{\|u\|_p = 1} f^\top u. \tag{8}$$

*Proof.* The proof can be found in [80]. $\qquad \square$

**Lemma B.4.** *For any $x, y \in \mathbb{R}^d$ and radii $r, s \geqslant 0$, the Minkowski sum of the two $\ell_p$-norm balls ($p \geqslant 1$)*

$$B_p(x, r) = \{u \in \mathbb{R}^d \mid \|u - x\| \leqslant r\}, \qquad B_p(y, s) = \{v \in \mathbb{R}^d \mid \|v - y\| \leqslant s\}$$

*is again a ball, namely*

$$B_p(x, r) + B_p(y, s) = B_p\big(x + y, r + s\big).$$

*Proof.* For convenience, we omit $p$ at the subscript in this proof. It suffices to show two inclusions.

- $B(x, r) + B(y, s) \subseteq B(x + y, r + s)$.

  Take any

  $$z = u + v, \quad u \in B(x, r), v \in B(y, s).$$

  Then by the triangle inequality,

  $$\big\| z - (x + y) \big\| = \big\| (u - x) + (v - y) \big\| \leqslant \|u - x\| + \|v - y\| \leqslant r + s.$$

  Hence $z \in B(x + y, r + s)$, proving the first inclusion.

- $B(x + y, r + s) \subseteq B(x, r) + B(y, s)$.

  Let $z \in B(x + y, r + s)$, so $\|z - (x + y)\| \leqslant r + s$. Set

  $$\alpha = \frac{r}{r + s}, \quad \beta = \frac{s}{r + s} \quad \text{(if } r + s = 0 \text{ then } r = s = 0 \text{ and the statement is trivial)}.$$

  Define $u = x + \alpha\big(z - (x + y)\big)$ and $v = y + \beta\big(z - (x + y)\big)$. Then

  $$u + v = x + y + (\alpha + \beta)\big(z - (x + y)\big) = x + y + z - (x + y) = z,$$

  and

  $$\|u - x\| = \alpha \|z - (x + y)\| \leqslant \frac{r}{r + s}(r + s) = r,$$

  $$\|v - y\| = \beta \|z - (x + y)\| \leqslant \frac{s}{r + s}(r + s) = s.$$

  Thus $u \in B(x, r)$ and $v \in B(y, s)$, so $z = u + v \in B(x, r) + B(y, s)$. This proves the reverse inclusion.

Combining (1) and (2) gives the desired equality $B(x, r) + B(y, s) = B\big(x + y, r + s\big)$. $\qquad \square$

**Lemma B.5.** *Let $g(u) = \sum_{i=1}^{N} \|u - u_i\|_p$. Then the Fenchel conjugate of $g : \mathbb{R}^d \to \mathbb{R}$ is*

$$g^*(u) = \begin{cases} \inf_{\substack{\sum_i y_i = y \\ \|y_i\|_q \leqslant 1}} \sum_i y_i^\top u_i, & \|y\|_q \leqslant 1, \\ +\infty, & \text{otherwise.} \end{cases}.$$

*Proof.* The key is that $g(u) = \sum_{i=1}^{N} \|u - u_i\|_p$ is a sum of $N$ "shifted-norms," and the Fenchel conjugate (Definition B.2) of a sum is the infimal convolution of the conjugates. We proceed in two steps.

- Let $f(x) = \|x\|_p$ be a single $\ell_p$-norm mapping and $q$ be the dual of $q$ satisfying $\frac{1}{p} + \frac{1}{q} = 1$ By [79], it is standard that

$$f^*(y) = \sup_{x \in \mathbb{R}^d} \{ y^\top x - \|x\|_p \} = \begin{cases} 0, & \|y\|_q \leqslant 1, \\ +\infty, & \text{otherwise.} \end{cases}$$

  Then we consider its "shift" by $u_i$. Let $f_i(u) := \|u - u_i\|_p$. By the translation rule for Fenchel conjugates [81, 82],

$$f_i^*(y) = \sup_u \{ y^\top u - \|u - u_i\|_p \} = \underbrace{\sup_t \{ y^\top (t + u_i) - \|t\|_p \}}_{t = u - u_i} = y^\top u_i + f^*(y).$$

  Hence, $f_i^*(y) = \begin{cases} y^\top u_i, & \|y\|_q \leqslant 1, \\ +\infty, & \text{otherwise.} \end{cases}.$

- As the Fenchel conjugate of $\|u - u_i\|_p$ has been evaluated, the Fenchel conjugate of their sum is given by

$$g^*(y) = \begin{cases} \inf_{\substack{\sum_i y_i = y \\ \|y_i\|_q \leqslant 1}} \sum_i y_i^\top u_i, & \|y\|_q \leqslant 1, \\ +\infty, & \text{otherwise.} \end{cases}.$$

  Applying this infimal convolution requires each component $f_i : \mathbb{R}^d \to \mathbb{R}$ is proper, convex, and lower semicontinuous, which is automatically satisfied by the $\ell_p$-norm.

$\square$

**Lemma B.6.** *Let $p_o, \lambda_o, \tilde{\omega}_o \in \mathbb{R}^d$ and $\gamma_o \geqslant 0$ be given constants. Let $\frac{1}{p} + \frac{1}{q} = 1$. Then the optimization problem*

$$\vartheta_o = \inf_{w : \|w\|_q \leqslant C} (p_o + \lambda_o)^\top w$$

*has the unique minimizer given by*

$$\begin{aligned} w_* &= -C \frac{J_q(p_o + \lambda_o)}{\|p_o + \lambda_o\|_p} \\ &= -C \frac{\text{sign}(p_o + \lambda_o) \odot |p_o + \lambda_o|^{p-1}}{\|p_o + \lambda_o\|_p^{p-1}}, \end{aligned}$$

*where $\odot$ represent the coordinate-wise product and $|\cdot|$ is the coordinate-wise absolute value. The optimal value is solved as*

$$\vartheta_o = -C \|p_o + \lambda_o\|_p.$$

*Proof.* By Hölder's inequality,

$$(p_o + \lambda_o)^\top w \geqslant -\|p_o + \lambda_o\|_p \|w\|_q.$$

To make the Hölder's inequality achieve the equality, we choose $w = -t J_q(p_o + \lambda_o)$ for some $t$, where $J_q$ is the $q$-unit vector defined in Eq. (8). Then by letting $\|w\|_q = C$, we obtain the final result. $\square$

**Lemma B.7.** *Suppose that $v \in \mathbb{R}^d$, $\{u_i\}_{i=1}^N \subset \mathbb{R}^d$, and $\mu \geqslant 0$, and the norm exponent $\frac{1}{p} + \frac{1}{q} = 1$ (for $p \geqslant 1$). Let $w = v + \mu\mathbf{1}$ and $C := \|w\|_q$. Then the optimization problem*

$$\min_{\{w_i\}_{i=1}^N \subset \mathbb{R}^d} \quad \sum_{i=1}^N w_i^\top u_i$$

$$s.t. \quad \sum_{i=1}^N w_i = w$$

$$\|w_i\|_q \leqslant C$$

*is feasible and solves the minimizer*

$$w_{i,*} = -\|v + \mu\mathbf{1}\|_q \frac{\mathrm{sign}(u_i + \tilde\lambda) \odot |u_i + \tilde\lambda|^{p-1}}{\|u_i + \tilde\lambda\|_p^{p-1}},$$

*for $i = 1, 2, \ldots, N$, where $\tilde\lambda^* \in \mathbb{R}^d$ is given by*

$$\tilde\lambda^* = \arg\min_{\tilde\lambda} \left\{ \tilde\lambda^\top w + \|w\|_q \sum_{i=1}^N \|u_i + \tilde\lambda\|_p \right\}. \tag{$*$}$$

*Proof.* We consider the constrained Lagrangian function

$$\tilde L(\{w_i\}, \tilde\lambda) = \sum_{i=1}^N \left[ u_i^\top w_i + \tilde\lambda^\top w_i \right] - \tilde\lambda^\top w.$$

where $\|w_i\|_q \leqslant C := \|w\|_q$. Let the dual function $\vartheta(\tilde\lambda) := \inf_{\{w_i\}} \tilde L(\{w_i\}, \tilde\lambda)$. Define

$$\vartheta_i(\tilde\lambda) = \inf_{w_i} \left( u_i + \tilde\lambda \right)^\top w_i.$$

By [Lemma B.6](), it solves

$$w_{i,*} = -C \frac{J_q(u_i + \lambda)}{\|u_i + \lambda\|_p}$$

$$= -C \frac{\mathrm{sign}(u_i + \tilde\lambda) \odot |u_i + \tilde\lambda|^{p-1}}{\|u_i + \tilde\lambda\|_p^{p-1}}$$

As the result,

$$\max_{\tilde\lambda} \vartheta(\tilde\lambda) = \min_{\tilde\lambda} \left\{ \tilde\lambda^\top w + C \sum_{i=1}^N \|u_i + \tilde\lambda\|_p \right\}$$

Recall that $w = v + \mu\mathbf{1} \in \mathbb{R}^d$ is a given vector. Denote

$$\tilde\lambda^* = \arg\min_{\tilde\lambda} \left\{ \tilde\lambda^\top w + C \sum_{i=1}^N \|u_i + \tilde\lambda\|_p \right\}.$$

Put it back to $w_{i,*}$, we obtain the final result. $\qquad\square$

**Lemma B.8.** *Suppose that $\{u_i\}_{i=1}^N \subset \mathbb{R}^d$, $\beta \geqslant 0$, and $v \in \mathbb{R}^d$. Let*

$$\varphi(\lambda, \mu) = -\lambda\beta + \inf_u \left[ (v + \mu\mathbf{1})^\top u + \lambda \sum_{i=1}^N \|u - u_i\|_p \right],$$

*where $p \in [1, +\infty]$ and $\frac{1}{p} + \frac{1}{q} = 1$. Then*

$$\sup_{\lambda \geqslant 0} \varphi(\lambda, \mu) = -\|v + \mu\mathbf{1}\|_q \beta + \sum_{i=1}^N w_{i,*}^\top u_i,$$

*where*

$$w_{i,*} := -\|v + \mu\mathbf{1}\| \frac{\text{sign}(u_i + \tilde{\lambda}_*) \odot |u_i + \tilde{\lambda}_*|^{p-1}}{\|u_i + \tilde{\lambda}_*\|_p^{p-1}}, \quad \tilde{\lambda}_* = \arg\min_{\tilde{\lambda}} \left\{ \tilde{\lambda}^\top w + C \sum_{i=1}^N \|u_i + \tilde{\lambda}\|_p \right\}.$$

*Consequently, the optimal $(\lambda^*, \mu^*)$ of achieving $\sup_{\lambda,\mu} \varphi(\lambda, \mu)$ is given by*

$$\mu^* = \arg\max_{\mu} \left\{ -\|v + \mu\mathbf{1}\|_q \beta + \sum_{i=1}^N w_{i,*}^\top u_i. \right\}, \quad and \quad \lambda^* = \|v + \mu^*\mathbf{1}\|_q.$$

*Proof.* We take the transformation $w := v + \mu\mathbf{1}$. Then

$$\varphi(\lambda, \mu) = -\lambda\beta + \inf_u \left[ w^\top u + \lambda \sum_{i=1}^N \|u - u_i\|_p \right].$$

For any convex $g$, by the definition of Fenchel conjugate (Definition B.2), we have

$$\inf_u \left[ w^\top u + \lambda g(u) \right] = -\lambda \underbrace{g^*(-w/\lambda)}_{= \sup_y \{ (-w/\lambda)^\top y - g(y) \}}.$$

Here $g(u) = \sum_i \|u - u_i\|_p$. Its conjugate is given by Lemma B.5,

$$g^*(z) = \begin{cases} \inf_{\sum_i z_i = z, \|z_i\|_q \leqslant 1} \sum_i z_i^\top u_i, & \|z\|_q \leqslant 1, \\ +\infty, & \text{otherwise.} \end{cases}$$

where $\frac{1}{p} + \frac{1}{q} = 1$. We put it back to $\inf_u \left[ w^\top u + \lambda g(u) \right] = -\lambda g^*(-w/\lambda)$, which leads to

$$\inf_u \left[ w^\top u + \lambda g(u) \right] = -\lambda \inf_{\substack{\sum_i z_i = -w/\lambda \\ \|z_i\|_q \leqslant 1}} \sum_i z_i^\top u_i,$$

where $\|\frac{w}{\lambda}\|_q \leqslant 1$. As the result, we take $w_i = -\lambda z_i$ to obtain

$$\inf_u \left[ w^\top u + \lambda g(u) \right] = \begin{cases} \inf_{\sum_i w_i = w, \|w_i\|_q \leqslant \lambda} \sum_{i=1}^N w_i^\top u_i, & \|w\|_q \leqslant \lambda, \\ -\infty, & \text{else.} \end{cases}$$

Thus, the full dual becomes

$$\varphi(\lambda, \mu) = \begin{cases} -\lambda\beta + \inf_{\sum_i w_i = v + \mu\mathbf{1}, \|w_i\|_q \leqslant \lambda} \sum_{i=1}^N w_i^\top u_i, & \|w\|_q \leqslant \lambda, \\ -\infty, & \text{else.} \end{cases}$$

For a fixed $\mu$, we need $\lambda \geqslant \|w\|_q$ to keep $\varphi$ finite, and $\varphi(\lambda, \mu)$ is decreasing in $\lambda$. Hence the best choice is

$$\lambda^* = \|w\|_q = \|v + \mu\mathbf{1}\|_q,$$

giving

$$\sup_{\lambda \geqslant 0} \varphi(\lambda, \mu) = -\|v + \mu\mathbf{1}\|_q \beta + \inf_{\sum_i w_i = v + \mu\mathbf{1}, \|w_i\|_q \leqslant \|v + \mu\mathbf{1}\|_q} \sum_{i=1}^N w_i^\top u_i.$$

It leads to another optimization problem $\inf_{\sum_i w_i = v + \mu\mathbf{1}, \|w_i\|_q \leqslant \|v + \mu\mathbf{1}\|_q} \sum_{i=1}^N w_i^\top u_i$. We construct another Lagrangian function to solve it. Denote $w_{i,*}$ as the minimizer given by Lemma B.7. Then we obtain

$$\max_{\lambda \geqslant 0} \varphi(\lambda, \mu) = -\|v + \mu\mathbf{1}\|_q \beta + \sum_{i=1}^N w_{i,*}^\top u_i.$$

It recovers the optimal dual variable $z$ is given by

$$z_* = -\sum_{i=1}^N \frac{\omega_{i,*}}{\lambda^*} = \sum_{i=1}^N \frac{\text{sign}(u_i + \tilde{\lambda}_*) \odot |u_i + \tilde{\lambda}_*|^{p-1}}{\|u_i + \tilde{\lambda}_*\|_p^{p-1}},$$

where

$$\tilde{\lambda}^* = \arg\min_{\tilde{\lambda}} \left\{ \tilde{\lambda}^\top w + C \sum_{i=1}^N \|u_i + \tilde{\lambda}\|_p \right\}.$$

□

**Lemma B.9.** *Suppose that $\{u_i\}_{i=1}^2 \subset \mathbb{R}$, $\lambda > 0$, $\beta \geqslant 0$, $w \in \mathbb{R}$ is non-zero, $q \in [1, +\infty]$, and $\frac{1}{p} + \frac{1}{q} = 1$. Define*

$$\varphi = \min_{u \in \mathbb{R}} \left\{ uw + \lambda \sum_{i=1}^2 |u - u_i| \right\}.$$

*Then when $|w| \leqslant 2\lambda$, the problem is solved as*

$$\varphi = w \frac{u_1 + u_2}{2} + (\lambda - \frac{|w|}{2})|u_1 - u_2|.$$

*Proof.* We start from the general case. Define $f(u) = uw + \lambda \sum_{i=1}^N |u - u_i|$. If $|w| \leqslant \lambda N$, then the sub-gradient is given by

$$\partial f(u) \ni w + \lambda \sum_{i=1}^N \text{sign}(u - u_i)$$

where $\text{sign}(t) := \begin{cases} -1, & t < 0, \\ 0, & t = 0, \\ 1, & t > 0. \end{cases}$    Write $\{u_i\}_{i=1}^N \subset \mathbb{R}$ in the increasing order:

$$u_{(1)} \leqslant u_{(2)} \leqslant \cdots \leqslant u_{(N)}.$$

Define $k^* := \lceil \frac{N - w/\lambda}{2} \rceil$. Then

$$u^* = u_{(k*)}$$

is the explicit solution. To prove it, we consider $u \in (u_{(k*)}, u_{(k*+1)})$; it is larger than exactly $k^*$ $u_i$'s. That is,

$$\partial f(u) = w + \lambda(k^* - (N - k^*)) \geqslant 0.$$

Whenever $u < u_{(k*)}$, the sign of sub-gradient becomes negative. As the result, $f(u)$ is decreasing when $u < u_{(k*)}$ then increasing when $u > u_{(k*)}$. Now we set $N = 2$. The problem gives

$$u^* = \frac{u_1 + u_2 - \text{sign}(w)|u_1 - u_2|}{2}.$$

When $w \geqslant 0$, $\text{sign}(w) = +1$ and

$$u^* = \frac{u_1 + u_2 - |u_1 - u_2|}{2} = \min\{u_1, u_2\} = u_{(1)}.$$

When $w < 0$, $\text{sign}(w) = -1$ and

$$u^* = \frac{u_1 + u_2 + |u_1 - u_2|}{2} = \max\{u_1, u_2\} = u_{(2)}.$$

Therefore, this formula recovers the original general case solution. Putting it back to $\varphi$ solves this problem. $\square$

The following lemma connects the sum-of-distance description to the quadratic form of an ellipse.

**Lemma B.10.** *Suppose that $\{u_i\}_{i=1}^2 \subset \mathbb{R}^d$, $\beta \geqslant 0$, and $\beta > \|u_1 - u_2\|_2$. The ellipse set is given by*

$$\mathcal{E} := \{u \mid \|u - u_1\|_2 + \|u - u_2\|_2 \leqslant \beta\}.$$

*Then there exists a matrix $A$ such that*

$$\mathcal{E} = \{u \mid (u - \bar{u})^\top A(u - \bar{u}) \leqslant 1\},$$

*where $\bar{u} := \frac{u_1 + u_2}{2}$. More explicitly, the matrix $A$ has the form*

$$A = \frac{4}{\beta^2 - \|u_2 - u_1\|_2^2} I - \frac{4}{\beta^2(\beta^2 - \|u_2 - u_1\|_2^2)}(u_2 - u_1)(u_2 - u_1)^\top.$$

*Proof.* Define

$$\bar{u} = \frac{u_1 + u_2}{2}, \quad f = \frac{\|u_2 - u_1\|_2}{2}, \quad a = \frac{\beta}{2}, \quad b = \sqrt{a^2 - f^2}, \quad e = \frac{u_2 - u_1}{\|u_2 - u_1\|_2},$$

and decompose each $u \in \mathbb{R}^d$ by

$$x = u - \bar{u}.$$

Then $u_1 = \bar{u} - fe$, $u_2 = \bar{u} + fe$, and

$$\|u - u_1\|_2 + \|u - u_2\|_2 = \|x + fe\|_2 + \|x - fe\|_2.$$

Hence

$$\|u - u_1\|_2 + \|u - u_2\|_2 \leqslant \beta \quad \Longleftrightarrow \quad \|x + fe\|_2 + \|x - fe\|_2 \leqslant 2a.$$

Now decompose $x$ into

$$\alpha = e^\top x, \qquad \xi^2 = \|x\|_2^2 - \alpha^2,$$

so that

$$\|x \pm fe\|_2 = \sqrt{(\alpha \pm f)^2 + \xi^2}.$$

The inequality $\sqrt{(\alpha + f)^2 + \xi^2} + \sqrt{(\alpha - f)^2 + \xi^2} \leqslant 2a$ is equivalent, after two squarings, to

$$\frac{\alpha^2}{a^2} + \frac{\xi^2}{b^2} \leqslant 1, \quad \text{where } b^2 = a^2 - f^2.$$

Finally, observe that

$$\alpha^2 = x^\top (ee^\top) x, \qquad \xi^2 = x^\top \left( I - ee^\top \right) x,$$

so

$$\frac{\alpha^2}{a^2} + \frac{\xi^2}{b^2} = x^\top \left( \tfrac{1}{a^2} ee^\top + \tfrac{1}{b^2} (I - ee^\top) \right) x.$$

Setting $A = \frac{1}{a^2} ee^\top + \frac{1}{b^2} (I - ee^\top)$ implies $(u - \bar{u})^\top A (u - \bar{u}) \leqslant 1$. It exactly characterizes $\{u \mid \|u - u_1\|_2 + \|u - u_2\|_2 \leqslant \beta\}$. Simplifying the form of $A$ leads to

$$A = \frac{4}{\beta^2 - \|u_2 - u_1\|_2^2} I - \frac{4}{\beta^2(\beta^2 - \|u_2 - u_1\|_2^2)} (u_2 - u_1)(u_2 - u_1)^\top.$$

This completes the proof. $\qquad\qquad\square$

**Lemma B.11.** *Let $v \in \mathbb{R}^d$, let $A \in \mathbb{R}^{d \times d}$ be symmetric positive definite, and let $\bar{p} \in \mathbb{R}^d$. Define*

$$F(\mu) = -\sqrt{(v + \mu\mathbf{1})^\top A^{-1} (v + \mu\mathbf{1})} + (v + \mu\mathbf{1})^\top \bar{p}, \quad \mu \in \mathbb{R}.$$

*Further set*

$$\alpha = \mathbf{1}^\top A^{-1} \mathbf{1}, \quad \beta = v^\top A^{-1} \mathbf{1}, \quad \gamma = v^\top A^{-1} v, \quad \delta = \mathbf{1}^\top \bar{p}.$$

*If $\delta^2 < \alpha$, then $F$ attains a unique maximizer*

$$\mu^* = -\frac{\beta}{\alpha} + \frac{\delta}{\alpha\sqrt{\alpha - \delta^2}} \sqrt{\alpha\gamma - \beta^2} = \mu^*$$

$$= -\frac{v^\top A^{-1} \mathbf{1}}{\mathbf{1}^\top A^{-1} \mathbf{1}} + \frac{\mathbf{1}^\top \bar{p}}{\mathbf{1}^\top A^{-1} \mathbf{1} \sqrt{\mathbf{1}^\top A^{-1} \mathbf{1} - (\mathbf{1}^\top \bar{p})^2}} \sqrt{(\mathbf{1}^\top A^{-1} \mathbf{1})(v^\top A^{-1} v) - (v^\top A^{-1} \mathbf{1})^2}.$$

*Proof.* We begin by computing the derivative of

$$F(\mu) = -\sqrt{(v + \mu\mathbf{1})^\top A^{-1} (v + \mu\mathbf{1})} + (v + \mu\mathbf{1})^\top \bar{p}.$$

Using the notation $\alpha = \mathbf{1}^\top A^{-1} \mathbf{1}$, $\beta = v^\top A^{-1} \mathbf{1}$, $\gamma = v^\top A^{-1} v$, and $\delta = \mathbf{1}^\top \bar{p}$, one checks

$$(v + \mu\mathbf{1})^\top A^{-1} (v + \mu\mathbf{1}) = \alpha\mu^2 + 2\beta\mu + \gamma,$$

so

$$F'(\mu) = -\frac{\alpha\mu + \beta}{\sqrt{\alpha\mu^2 + 2\beta\mu + \gamma}} + \delta.$$

Setting $F'(\mu) = 0$ gives the stationarity condition

$$\alpha\mu + \beta = \delta\sqrt{\alpha\mu^2 + 2\beta\mu + \gamma},$$

which upon squaring yields the quadratic equation

$$(\alpha^2 - \alpha\delta^2)\mu^2 + 2\beta(\alpha - \delta^2)\mu + (\beta^2 - \delta^2\gamma) = 0.$$

Let $\delta^2 < \alpha$. Here $\alpha - \delta^2 > 0$, so dividing by $\alpha - \delta^2$ gives

$$\alpha\mu^2 + 2\beta\mu + \frac{\beta^2 - \delta^2\gamma}{\alpha - \delta^2} = 0,$$

whose two roots are

$$\mu = -\frac{\beta}{\alpha} \pm \frac{\delta}{\alpha\sqrt{\alpha - \delta^2}}\sqrt{\alpha\gamma - \beta^2}.$$

One verifies by inspecting $\lim_{\mu\to\pm\infty} F'(\mu) = \delta \mp \sqrt{\alpha}$ that exactly the "+" choice yields a change of sign from + to −, and hence is the unique global maximizer. $\qquad\square$

## B.2 Implicit Solution

In this subsection, we recap and prove the full version of Theorem 3.4.

**Theorem B.12.** *Let $d := |\mathcal{S}|$ be the cardinal of state space and $\frac{1}{p} + \frac{1}{q} = 1$ for $p, q \in [1, +\infty]$. Suppose that $\{u_i\}_{i=1}^N \subset \mathbb{R}^d$ for each $(s, a) \in \mathcal{S} \times \mathcal{A}$, the uncertainty size $\beta \geqslant 0$, and $v \in \mathbb{R}^d$. The solution of*

$$\begin{aligned} \min_{u\in\mathbb{R}^d} \quad & v^\top u \\ s.t. \quad & \sum_{i=1}^N \|u - u_i\|_p \leqslant \beta, \\ & \mathbf{1}^\top u = 0. \end{aligned} \tag{9}$$

*is given by*

$$u^* = \arg\min_u \left[(v + \mu^*\mathbf{1})^\top u + \lambda^* \sum_{i=1}^N \|u - u_i\|_p\right],$$

*where*

$$\begin{cases} \mu^* = \arg\max_\mu \left\{-\|v + \mu\mathbf{1}\|_q\beta + \sum_{i=1}^N w_{i,*}^\top u_i.\right\}, \\ \lambda^* = \|v + \mu^*\mathbf{1}\|_q, \\ \tilde{\lambda}_*(\mu) = \arg\min_{\tilde{\lambda}} \left\{\tilde{\lambda}^\top(v + \mu\mathbf{1}) + \|v + \mu\mathbf{1}\|_q \sum_{i=1}^N \|u_i + \tilde{\lambda}\|_p\right\}, \\ w_{i,*}(\mu) = -\|v + \mu\mathbf{1}\|_q \frac{\operatorname{sign}(u_i+\tilde{\lambda}_*)\odot|u_i+\tilde{\lambda}_*|^{p-1}}{\|u_i+\tilde{\lambda}_*\|_p^{p-1}}, \end{cases} \tag{10}$$

*Remark* (The procedure of solving $u^*$). To obtain $u^*$, it suffices to solve $\mu^*$ and $\lambda^*$ as $v$ and all $u_i$'s have been given. The first step is to solve

$$\begin{cases} \tilde{\lambda}_*(\mu) = \arg\min_{\tilde{\lambda}} \left\{\tilde{\lambda}^\top(v + \mu\mathbf{1}) + \|v + \mu\mathbf{1}\|_q \sum_{i=1}^N \|u_i + \tilde{\lambda}\|_p\right\}. \\ w_{i,*}(\mu) = -\|v + \mu\mathbf{1}\|_q \frac{\operatorname{sign}(u_i+\tilde{\lambda}_*)\odot|u_i+\tilde{\lambda}_*|^{p-1}}{\|u_i+\tilde{\lambda}_*\|_p^{p-1}}, \end{cases}$$

for $i = 1, 2, \ldots, N$. Both variables depend on the variable $\mu$ and other values are known. The next step is to solve

$$\begin{cases} \mu^* = \arg\max_\mu \left\{-\|v + \mu\mathbf{1}\|_q\beta + \sum_{i=1}^N w_{i,*}^\top u_i.\right\}, \\ \lambda^* = \|v + \mu^*\mathbf{1}\|_q. \end{cases}$$

Once $(\mu^*, \lambda^*)$ is solved, the primal variable $\mu^*$ is obtained immediately.

*Proof.* Our goal is to solve the following constrained optimization problem:

$$\min_{u \in \mathbb{R}^d} \quad v^\top u$$

$$\text{s.t.} \quad \sum_{i=1}^N \|u - u_i\|_p \leqslant \beta,$$

$$\mathbf{1}^\top u = 0.$$

As it is a constrained optimization problem, the standard approach of solving this problem is using Lagrangian multipliers. we introduce the Lagrangian multipliers $\lambda \geqslant 0$ for the inequality and $\mu \in \mathbb{R}$ for the equality. The Lagrangian function is

$$L(u, \lambda, \mu) = v^\top u + \lambda \Big( \sum_{i=1}^N \|u - u_i\|_p - \beta \Big) + \mu(\mathbf{1}^\top u),$$

with $\lambda \geqslant 0$, $\mu \in \mathbb{R}$. Because this optimization problem is a standard convex optimization problem with satisfying the Slater's condition, we have the strong duality

$$\underbrace{\inf_u \sup_{\lambda, \mu} L(u, \lambda, \mu)}_{\text{original opt. prob.}} = \sup_{\lambda, \mu} \inf_u L(u, \lambda, \mu).$$

Then we turn the original optimization problem into solving its dual-form problem. We let the dual function be $\varphi(\lambda, \mu) := \inf_u L(u, \lambda, \mu)$. Then

$$\varphi(\lambda, \mu) = -\lambda\beta + \inf_u \Big[ (v + \mu\mathbf{1})^\top u + \lambda \sum_{i=1}^N \|u - u_i\|_p \Big].$$

The above formulation plays the crucial role in our proof. For the implicit solution, we will follow [Lemma B.8](#) to complete the remaining calculation. For the explicit solution, this dual form can be significantly simplified in some cases.

By [Lemma B.8](#), the dual form can be simplified as

$$\max_{\lambda, \mu} \varphi(\lambda, \mu) = -\|v + \mu^*\mathbf{1}\|_q \beta + \sum_{i=1}^N w_{i,*}^\top u_i, .$$

where

$$w_{i,*} := -\|v + \mu\mathbf{1}\| \frac{\text{sign}(u_i + \tilde{\lambda}_*) \odot |u_i + \tilde{\lambda}_*|^{p-1}}{\|u_i + \tilde{\lambda}_*\|_p^{p-1}}, \quad \tilde{\lambda}_* = \arg\min_{\tilde{\lambda}} \Big\{ \tilde{\lambda}^\top w + C \sum_{i=1}^N \|u_i + \tilde{\lambda}\|_p \Big\}.$$

and

$$\mu^* = \arg\max_\mu \Big\{ -\|v + \mu\mathbf{1}\|_q \beta + \sum_{i=1}^N w_{i,*}^\top u_i. \Big\}, \quad \lambda^* = \|v + \mu^*\mathbf{1}\|_q.$$

The optimal primary variable is given as

$$u^* = \arg\min_u \Big[ (v + \mu^*\mathbf{1})^\top u + \lambda^* \sum_{i=1}^N \|u - u_i\|_p \Big].$$

$\square$

## B.3 Explicit Solution

In this subsection, we recap and prove the full version of [Theorem 3.5](#).

**Theorem B.13.** *Let $d := |\mathcal{S}|$ be the cardinal of state space. Suppose that $\{u_i\}_{i=1}^N \subset \mathbb{R}^d$ for each $(s, a) \in \mathcal{S} \times \mathcal{A}$, the uncertainty size $\beta \geqslant 0$, and $v \in \mathbb{R}^d$. The optimization problem defined by [Eq. (4)](#) is explicitly solved in the following cases:*

*(a) Let $N = 1$. then*

$$\mu^* = u_1 + \beta \frac{\mathrm{sign}(v + \mu^*\mathbf{1}) \odot |v + \mu^*\mathbf{1}|^{q-1}}{\|v + \mu^*\mathbf{1}\|_q^{q-1}}.$$

*(b) Let $p = 1$ and $N = 2$. Define $\bar{u} = \frac{u_1 + u_2}{2}$,*

$$\mu^* = -\frac{v_{\max} + v_{\min}}{2} \quad \text{and} \quad \lambda^* = \frac{1}{2}\|v + \mathbf{1}\mu^*\|_\infty = \frac{v_{\max} - v_{\min}}{4}.$$

*Suppose that $\beta > \|u_2 - u_1\|_1$. Then the explicit solution to the optimization problem Eq. (4) is given as*

$$u^* = \bar{u} - \frac{\beta - \|u_1 - u_2\|_1}{2} \left[ \mathrm{sign}(v + \mu^*\mathbf{1}) \odot \mathbb{I}_{\{|v + \mu^*\mathbf{1}| = 2\lambda^*\}} \right].$$

*(c) Let $p = 2$ and $N = 2$. Define $\bar{u} = \frac{u_1 + u_2}{2}$,*

$$\Omega := \Omega(u_1, u_2, \beta) = \left[ I - \frac{1}{\beta^2}(u_2 - u_1)(u_2 - u_1)^\top \right], \tag{11}$$

$$\underline{\lambda}^* = \sqrt{(v + \mu^*\mathbf{1})^\top \Omega^{-1}(v + \mu^*\mathbf{1})}, \text{ and } \mu^* = -\frac{v^\top \Omega^{-1}\mathbf{1}}{\mathbf{1}^\top \Omega^{-1}\mathbf{1}}. \tag{12}$$

*Suppose that $\beta > \|u_2 - u_1\|_2$. Then the explicit solution to the optimization problem Eq. (4) is given as*

$$u^* = \bar{u} - \frac{\sqrt{\beta^2 - \|u_2 - u_1\|_2^2}}{2} \frac{1}{\underline{\lambda}^*} \Omega^{-1}(v + \mu^*\mathbf{1}).$$

*Proof.* We follow the standard routine used in proving Theorem B.12.

(a) When $N = 1$ the objective optimization problem is given by

$$\min_{u \in \mathbb{R}^d} \quad v^\top u$$
$$\text{s.t.} \quad \|u - u_1\|_p \leqslant 1,$$
$$\mathbf{1}^\top u = 0.$$

We take the transformation $u' = u - u_1$. It still satisfies $\mathbf{1}^\top u' = 0$. Then the problem become

$$\min_{u' \in \mathbb{R}^d} \quad v^\top u'$$
$$\text{s.t.} \quad \|u'\|_p \leqslant 1,$$
$$\mathbf{1}^\top u' = 0.$$

This transformation has turned this problem into the standard $\ell_p$-norm structure, which has been explicitly solved in [39, 40]. The optimal $u'$ is given for arbitrary $p \geqslant 1$ as

$$u'_* = \beta \frac{\mathrm{sign}(v + \mu^*\mathbf{1})|v + \mu^*\mathbf{1}|^{q-1}}{\|v + \mu^*\mathbf{1}\|_q^{q-1}},$$

where $\mu^* = \arg\min_{\mu \in \mathbb{R}} \|v + \mu\mathbf{1}\|_q$. As the result,

$$u^* = u'_* + u_1 = u_1 + \beta \frac{\mathrm{sign}(v + \mu^*\mathbf{1})|v + \mu^*\mathbf{1}|^{q-1}}{\|v + \mu^*\mathbf{1}\|_q^{q-1}}.$$

(b) When $p = 1$, we define the order

$$u_{(1),j} \leqslant u_{(2),j} \leqslant \cdots \leqslant u_{(N),j},$$

We follow the same routine as Theorem B.12 and derive the dual function $\varphi(\lambda, \mu)$:

$$\varphi(\lambda, \mu) = -\lambda\beta + \inf_u \left[ (v + \mu\mathbf{1})^\top u + \lambda \sum_{i=1}^N \|u - u_i\|_1 \right]$$

$$\overset{(i)}{=} -\lambda\beta + \inf_u \left[ \sum_{j=1}^d (v_j + \mu)u_j + \lambda \sum_{i=1}^N \sum_{j=1}^d |u_j - u_{ij}| \right]$$

$$= -\lambda\beta + \sum_{j=1}^d \inf_{u_j} \left[ (v_j + \mu)u_j + \lambda \sum_{i=1}^N |u_j - u_{ij}| \right].$$

where $(i)$ decomposes the $\ell_1$-norm by coordinates. When $N = 2$, by Lemma B.9, the dual function is solved as

$$\varphi(\lambda, \mu) = -\lambda\beta + \sum_{j=1}^d \inf_{u_j} \left[ (v_j + \mu)u_j + \lambda \sum_{i=1}^N |u_j - u_{ij}| \right]$$

$$= -\lambda\beta + \sum_{j=1}^d \left[ (v_j + \mu)\frac{u_{1j} + u_{2j}}{2} + (\lambda - \frac{|v_j + \mu|}{2})|u_{1j} - u_{2j}| \right]$$

$$= -\lambda\beta + (v + \mathbf{1}\mu)^\top \bar{u} + [2\lambda\mathbf{1} - (v + \mathbf{1}\mu)]^\top |\frac{u_1 - u_2}{2}|.$$

As the smaller $\lambda$ is, the larger $\varphi(\lambda, \mu)$ is. It achieves the supremum at $\lambda^* = \max_j \frac{v_j + \mu}{2} = \frac{1}{2}\|v + \mathbf{1}\mu\|_\infty$. Then we solve

$$\sup_\lambda \varphi(\lambda, \mu)$$

$$= -\frac{1}{2}\|v + \mathbf{1}\mu\|_\infty\beta + (v + \mathbf{1}\mu)^\top \bar{u} + \|v + \mathbf{1}\mu\|_\infty\|\frac{u_1 - u_2}{2}\|_1 - (v + \mathbf{1}\mu)^\top |\frac{u_1 - u_2}{2}|.$$

Then we have

$$\sup_\mu \sup_\lambda \varphi(\mu, \lambda) = -\frac{\beta - \|u_2 - u_1\|_1}{2} \inf_\mu \left[ \|v + \mu\mathbf{1}\|_\infty + \frac{(v + \mu\mathbf{1})^\top |u_1 - u_2|}{\beta - \|u_2 - u_1\|_1} \right] + v^\top \bar{u}.$$

As $\beta - \|u_2 - u_1\|_1 > 0$, it solves

$$\mu^* = -\frac{v_{\max} + v_{\min}}{2} \quad \text{and} \quad \lambda^* = \frac{1}{2}\|v + \mathbf{1}\mu^*\|_\infty = \frac{v_{\max} - v_{\min}}{4}.$$

Now we consider the KKT condition of the original Lagrangian function. We solve

$$\partial_{u_j} L(u, \lambda, \mu) = \lambda\,\text{sign}(u_j - u_{1j}) + \lambda\,\text{sign}(u_j - u_{2j}) + v_j + \mu \ni 0.$$

For inactive coordinate, the optimal value is attained for arbitrary $u_j \in (u_{(1)j}, u_{(2)j})$; in these cases, we simply take $u_j = \frac{u_{1j} + u_{1j}}{2}$. There are exactly two active coordinates matching the corner-case condition $|v_j + \mu^*| = 2\lambda^*$: $j^* = \arg\max v_j$ and $j_* = \arg\min v_j$. In these cases we take $u_j$ as $\frac{u_{1j} + u_{1j}}{2}$ subtracting a drift. It finally solves

$$u^* = \bar{u} - \frac{\beta - \|u_1 - u_2\|_1}{2} \left[ \text{sign}(v - \frac{v_{\max} + v_{\min}}{2}) \odot \mathbb{I}_{\{|v - \frac{v_{\max} + v_{\min}}{2}| = \frac{v_{\max} - v_{\min}}{2}\}} \right].$$

The magnitude coefficient is used to ensure that $u^*$ belongs to $\|u^*\|_1 \le \beta$.

(c) By Lemma B.10, there exists a semi-positive definite matrix $A$ such that

$$\{u \mid \|u - u_1\|_2 + \|u - u_2\|_2 \le \beta\} = \{u \mid (u - \bar{u})^\top A(u - \bar{u}) \le 1\},$$

where $\bar{u} := \frac{u_1 + u_2}{2}$. As the result, the objective optimization problem can be simplified as

$$\min_{u \in \mathbb{R}^d} \quad v^\top u$$

$$\text{s.t.} \quad (u - \bar{u})^\top A(u - \bar{u}) \le 1,$$

$$\mathbf{1}^\top u = 0.$$

We follow the same routine as Theorem B.12 and derive the dual function $\varphi(\lambda, \mu)$:

$$\varphi(\lambda, \mu) = -\lambda + \inf_u \left[ (v + \mu\mathbf{1})^\top u + \lambda(u - \bar{u})^\top A(u - \bar{u}) \right]$$
$$= -\lambda + (v + \mu\mathbf{1})^\top \bar{u} - \frac{1}{4\lambda}(v + \mu\mathbf{1})^\top A^{-1}(v + \mu\mathbf{1}),$$

where the infimum is attained at $u^* = \bar{u} - \frac{1}{2\lambda}A^{-1}(v + \mu\mathbf{1})$. Then

$$\sup_{\lambda, \mu} \varphi(\lambda, \mu) = \sup_{\lambda, \mu} \left\{ -\lambda + (v + \mu\mathbf{1})^\top \bar{u} - \frac{1}{4\lambda}(v + \mu\mathbf{1})^\top A^{-1}(v + \mu\mathbf{1}) \right\}$$
$$\overset{(i)}{=} \sup_\mu \left\{ -\sqrt{(v + \mu\mathbf{1})^\top A^{-1}(v + \mu\mathbf{1})} + (v + \mu\mathbf{1})^\top \bar{u} \right\}$$

where (i) applies the optimal choice $\lambda^*(\mu) = \frac{1}{2}\sqrt{(v + \mu\mathbf{1})^\top A^{-1}(v + \mu\mathbf{1})}$ and $\mu^*$ is given by Lemma B.11 to solve this maximization problem. As the result,

$$u^* = \bar{u} - \frac{1}{2\lambda^*}A^{-1}(v + \mu^*\mathbf{1})$$

where $\lambda^* = \frac{1}{2}\sqrt{(v + \mu^*\mathbf{1})^\top A^{-1}(v + \mu^*\mathbf{1})}$ and

$$\mu^* = -\frac{v^\top A^{-1}\mathbf{1}}{\mathbf{1}^\top A^{-1}\mathbf{1}} + \frac{\mathbf{1}^\top \bar{p}}{\mathbf{1}^\top A^{-1}\mathbf{1}\sqrt{\mathbf{1}^\top A^{-1}\mathbf{1} - (\mathbf{1}^\top \bar{p})^2}}\sqrt{(\mathbf{1}^\top A^{-1}\mathbf{1})(v^\top A^{-1}v) - (v^\top A^{-1}\mathbf{1})^2}.$$

Here, the matrix $A$ is determined by $\{u_1, u_2\}_{i=1}^2$ and the vector $v$ in Lemma B.10. We recall that $\mathbf{1}^\top \bar{u} = 0$. Therefore, $\mu^*$ is further simplified as

$$\mu^* = -\frac{v^\top A^{-1}\mathbf{1}}{\mathbf{1}^\top A^{-1}\mathbf{1}}.$$

It completes the proof in the part (c).

$\square$

# C Experiment Details

In this section, we include the omitted details of Section 4. All source codes and hyper-parameter settings are available in the supplementary materials.

## C.1 Hardware and System Environment

We conducted our experiments on a laptop running Windows 11 Home. The device is equipped with 32GB of RAM, 1TB SSD, an AMD Ryzen 9 7940HS processor and a NVIDIA GeForce RTX 4070 Laptop GPU. Our implementation was tested using Python version 3.10.10. Additional dependencies are listed in the supplementary `requirements.txt` file.

The actual hardware requirement for running our implementation is significantly lower than the specification listed above. Most experiments can be reproduced on consumer-grade machines with 8–16GB RAM and any CUDA-compatible NVIDIA GPU.

## C.2 Task Descriptions

**Multi-Asset Portfolio Rebalancing** This task captures the setting where an institutional investor reallocates capital across multiple assets over discrete time intervals to maintain a desired risk-return profile. It reflects strategic portfolio management, such as end-of-day rebalancing or tactical asset allocation. The task emphasizes robustness to market impact under varying capital scales, which is critical for large-volume institutional strategies. In our experiment, we select five representative ETFs, SPY, TLT, GLD, EFA, and VNQ, and use historical data from May 9, 2021 to May 9, 2022 for training. Evaluation is conducted on the out-of-sample period from June 9 to December 9, 2022.

Table 2: Example of ask-side execution from an AMZN order book snapshot (21 June 2012). This table shows how our simulator determines the execution price for a market buy order. Instead of using the last trade price, the simulator consumes liquidity by filling shares at each ask level in order, starting from the best price. It calculates the VWAP based on the prices and quantities filled, and uses this VWAP as the final executed price.

| Level | Price (USD) | Depth (sh) | Exec. 100 sh | Exec. 1 000 sh |
|---|---|---|---|---|
| Level 1 | 223.95 | 100 | 100 | 100 |
| Level 2 | 223.99 | 100 | 0 | 100 |
| Level 3 | 224.00 | 220 | 0 | 220 |
| Level 4 | 224.25 | 100 | 0 | 100 |
| Level 5 | 224.40 | 547 | 0 | 480 |
| **VWAP paid** | — | — | **223.95** | **224.21** |

**Single-Asset Intra-Day Trading**   This task models the decision-making process of an agent that repeatedly buys or sells a single asset over short time intervals, such as minutes or seconds. It reflects the setting of mid-frequency or algorithmic trading, where even small trades can shift market prices. The goal is to maximize the risk-adjusted cumulative return while accounting for execution slippage, making it a standard benchmark for evaluating RL-based trading strategies. We follow the exactly same training and evaluation period as the multi-asset portfolio rebalancing task. In our experiments, we use minute-level historical data for the assets including META, MSFT, and SPY. The observation includes stock price, trading volume, implied volatility, and current portfolio return. During evaluate, we reconstruct the LOB dynamics using the tick-level trade data; the execution price is then simulated as illustrated in Table 2. Same as the multi-asset portfolio rebalancing experiment, the training data covers the period from May 9th, 2021 to May 9th, 2022, and the evaluation period is from June 9 to December 9, 2022. All data are accessed via the Polygon.io Stock Market API.

### C.3   Reinforcement Learning Framework

We implemented a Gym-like RL trading environment [77, 78] using the historical data including the stock price, trading volume, the implied volatility, and the current portfolio return. Instead of using a single timestep data, we set `lookback_period` $= 30$ to account for $30$ previous timestep; as the result, the observed state contains `lookback_period` $\times 4 = 120$ dimension. Given the state described above, we assign the transition probability from the current state $s_t$ to the next state $s_{t+1}$ as the probability $1$, independent with the action. The action is implemented as a single float value that determines the position size relative to the maximum possible position size based on the available capital. The reward is a Sharpe-like ratio that consists of two components:

$$\texttt{reward} = \frac{\texttt{current\_return}}{\texttt{current\_volatility} + \varepsilon} - \delta \times \frac{|\texttt{current\_position} - \texttt{previous\_position}|}{\texttt{max\_shares}}.$$

where $\varepsilon > 0$ is a small constant to ensure numerical stability, and $\delta$ is a tunable coefficient controlling the strength of the transaction cost penalty. The first term encourages higher risk-adjusted return, while the second term discourages frequent or drastic position shifts, which aligns with practical trading considerations under market impact or slippage. We also include $0.1\%$ transaction cost throughout the experiment.

**Uncertainty Restriction to Execution Prices**   Instead of perturbing the whole transition probability, we restrict the perturbation on the execution price. The formulation of restriction is given as follows: we denote the state $s_t$ as two components $(p_t, f_t)$, and we hope perturb the transition on $p_t$ only. We re-write the transition probability using the conditional probability

$$\begin{aligned} \mathbb{P}(s_{t+1} \mid s_t, a_t) &= \mathbb{P}(p_{t+1}, f_{t+1} \mid s_t, a_t) \\ &= \mathbb{P}(p_{t+1} \mid s_t, a_t)\mathbb{P}(f_{t+1} \mid p_{t+1}, s_t, a_t). \end{aligned}$$

Then we set the uncertainty $u$ as the perturbation on the kernel $\mathbb{P}(p_{t+1} \mid s_t, a_t)$ instead of the whole transition kernel $\mathbb{P}(s_{t+1} \mid s_t, a_t)$.

**Momentum Strategy** The momentum strategy is a classical intra-day minute-level algorithmic trading method that exploits intraday time-series momentum to generate trading signals. We follow the classical implementation presented by [76]. The core idea of the momentum strategy is based on the empirical observation where assets that have performed well in the recent past are more likely to continue performing well in the near future, while poorly performing assets are likely to continue underperforming.

## C.4 Parameter Details

Detailed descriptions of all parameters are provided in the separate supplementary materials.

## C.5 Omitted Visualization

In Figure 3, we have visualized the return curve of the META stock. In this subsection, we include the visualization of the other two assets.

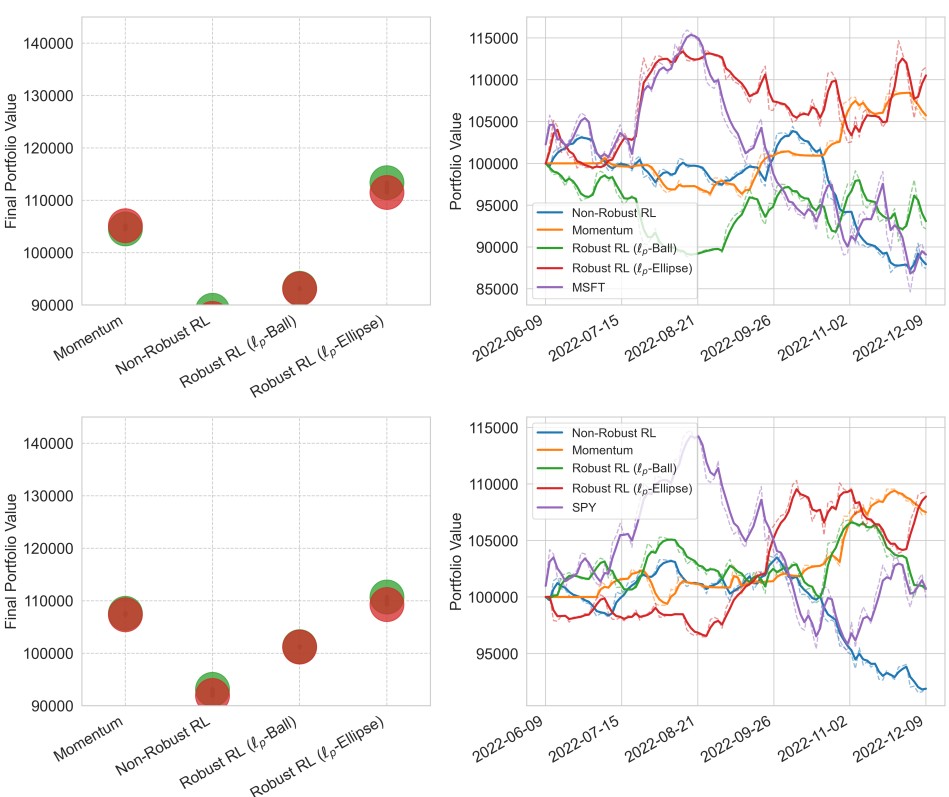

Figure 5: Performance comparison of trading strategies on the MSFT stock and SPY ETF.

## C.6 Other Implementation Details

In this subsection, we discuss the practical implementation considerations and provide the details for reproducing our experiments. The source codes are also provided in the supplementary material.

**RL Algorithm and Value Function Approximations** We use the standard Robust Actor-Critic algorithm [43] to train the RL agent with replacing its IPM uncertainty set or doubly-sampling uncertainty set with the $\ell_p$-ellipse uncertainty set and the $\ell_p$-norm uncertainty set. We adopt an Actor-Critic architecture with a shared feature extractor and separate heads for the actor and critic. The shared backbone consists of three fully connected layers with ReLU activations, mapping the (flattened) input state to a high-level representation. The actor head outputs the vector with the same dimension as the action through a two-layer MLP followed by a sigmoid/tanh activation, depending

on the underlying task. The critic head predicts the state value using a similar two-layer MLP. All linear layers are orthogonally initialized to promote stable training. When updating the Actor network, we apply the classical PPO algorithm [83].

**Discretization of Execution Prices**    To apply the robust RL framework, we apply the discretization to the execution price. We introduce two additional hyper-parameter to control the discretization level: $2N + 1$ represents the number of total discretization in the execution prices and $\delta$ represents the strength of each level. Given the execution price $p$, we have $2N + 1$ potential execution prices $[p - N\delta, \ldots, p - \delta, p, p + \delta, \ldots, p + N\delta]$ in total. This discretization approach allows us to directly apply the closed-form solution to solve the optimal $u^*$.

# D    Limitations

Despite the promising results, this work has several limitations. First, the theoretical analysis largely relies on the assumption of finite state and action spaces for tractability, which is primarily due to the limited development of deep learning theory. Second, the market impact simulation during the evaluation stage, while based on trade-level data, remains an approximation and may lack accuracy for extreme volumes. In fact, when the volume is sufficiently high, it often triggers momentum-based strategies deployed by other institutions in the market, resulting in higher market impact than reflected VWAP. Third, although our experiments demonstrate the robustness of the $\ell_p$-ellipse uncertainty set under increasing volumes, we do not test its robustness under increasing trading frequency. Designing experiments in this setting is more challenging, as the behavior of RL agents varies significantly across different time scales.

# E    Additional Supplementary Materials

In this additional supplementary material[3], we provide several components omitted in our previous technical appendix. It includes: (1) The theoretical analysis of the robust TD-learning. (2) The further breakdown of the parameter setting and the grid-search strategy to optimize the optimal parameter. (3) The other implementation details that are related to our robust reinforcement learning setting.

## E.1    Convergence of Robust TD-Learning

**Theorem** (Convergence of Robust TD-Learning). *If the uncertainty set is defined as the $\ell_p$-ellipse uncertainty set, then the worst-case transition probability $P_+(\cdot|s, a)$ can be represented as*

$$P_+(\cdot|s, a) = P_0(\cdot|s, a) + u^*$$

*where $\beta$ is the radius of the uncertainty set and $u^*$ is solved in Theorem 3.4 and Theorem 3.5.*

*Remark.* For non-robust policy evaluation, variance-reduction techniques are often employed to accelerate training [84–86]. For robust policy evaluation, the most recent progress can be found in [87]. For simplicity, we only present a naive robust TD-learning approach.

*Proof.* We apply the following TD-learning update rule:

$$V(s) \leftarrow V(s) + \eta \left( \underbrace{r(s, a) + \gamma V(s') - V(s)}_{\text{stand. TD err. under } P_0} + \gamma V^\top u^* \right). \tag{13}$$

---

[3]This section was originally submitted as a separate supplementary material. We include it here for the reader's convenience.

It is easy to observe that this update rule is equivalent to the non-robust TD-learning over the worst-case transition probability:

$$\mathbb{E}_{s' \sim P_0(s'|s,a)}[r(s,a) + \gamma V(s')] + \gamma \langle u^*, V \rangle$$
$$= r(s,a) + \gamma \sum_{s',a} P_0(s'|s,a)\pi(a|s)V(s') + \gamma \sum_{s'} u^*(s')V(s')$$
$$= r(s,a) + \gamma \sum_{s',a} P_0(s'|s,a)\pi(a|s)V(s') + \gamma \sum_{s',a} \pi(a|s)u^*(s')V(s')$$
$$= r(s,a) + \gamma \sum_{s',a} \pi(a|s)[P_0(s'|s,a) + u^*(s')]V(s')$$
$$\overset{(i)}{=} r(s,a) + \gamma \sum_{s',a} \pi(a|s)P_+(s'|s,a)V(s')$$
$$= r(s,a) + \gamma \mathbb{E}_{s' \sim P_+(s'|s,a)}V(s'),$$

where (i) applies the worst-case transition probability we have derived. Therefore, by applying existing TD-learning convergence analysis, we obtain that the convergence rate is also $\frac{1}{\sqrt{K}}$. □

## E.2 Parameter Setting and Grid-Search

We apply grid-search to select hyperparameters to balance learning performance, computational efficiency, and robustness to uncertainty. In this section, we detail the architectural and training hyperparameters used across our three model variants: standard RL, elliptic uncertainty robust RL, and ball-shaped uncertainty robust RL.

### E.2.1 Model Architecture

All models share the same actor-critic network architecture, which consists of:

- **Feature Extractor**: Three fully-connected layers with ReLU activation functions, each containing 256 neurons.
- **Actor Network**: Two fully-connected layers (256 and 128 neurons) with the final layer using a $\tanh$ activation function to bound actions within $[-1, 1]$.
- **Critic Network**: Two fully-connected layers (256 and 128 neurons) with a linear output layer.
- **Weight Initialization**: Orthogonal initialization with a gain of $\sqrt{2}$ for linear layers to improve training stability.

This architeture is selected by default and we did not further tune the model architecture.

### E.2.2 Training Parameters

We employ the Proximal Policy Optimization (PPO) algorithm with the following hyperparameters:

Table 3: PPO Training Hyperparameters. The PPO clip parameter $\epsilon$ regulates the range of policy updates to ensure stability. Policy update epochs indicate how many times each batch is reused for learning. Batch size is the number of trajectories used per update.

| Parameter | Standard RL | Robust RL (Elliptic) | Robust RL (Ball) |
|---|---|---|---|
| Learning Rate | $3 \times 10^{-4}$ | $3 \times 10^{-4}$ | $3 \times 10^{-4}$ |
| Discount Factor ($\gamma$) | 0.99 | 0.99 | 0.99 |
| PPO Clip Parameter ($\epsilon$) | 0.2 | 0.2 | 0.2 |
| Policy Update Epochs | 10 | 10 | 10 |
| Batch Size | 64 | 64 | 64 |

We apply the grid-search on the learning rate and the PPO clip parameter; however, the grid-search is not applied to improve the performance. Instead, we apply the grid-search to identify a default

parameter group to ensure the algorithm convergence. In the remaining experiments, we will use the same parameter in the robust RL method for fair comparison.

### E.2.3 Robustness Parameters

For our robust RL implementations, we include another group of hyper-parameters.

Table 4: Robustness Parameters. The robust type is the string used in our codes to determine the different types of uncertainty sets. The parameter $\beta$ is used to determine the size of uncertainty set.

| Parameter | Robust RL (Elliptic) | Robust RL (Ball) |
|---|---|---|
| Robust Type | P1N2 | P1 |
| Uncertainty Set Parameter ($\beta$) | $1 \times 10^{-4}$ | $1 \times 10^{-4}$ |
| Uncertainty Dimension | 3 | 3 |
| Epsilon | $1 \times 10^{-3}$ | $1 \times 10^{-3}$ |

We grid-search he parameter $\beta$ from $[0.1, 0.01, 0.001, 0.0001, 0.00001]$ and use the best result on the training period with adopting the early stopping. Then the best model is evaluated in the testing period to obtain the final performance.

### E.3 Other Implementation Details

The learning rate scheduler employs a ReduceLROnPlateau strategy with a factor of 0.5 and patience of 5 episodes. This adaptive approach reduces the learning rate when performance plateaus. Additionally, gradient clipping with a threshold of 0.5 is applied to prevent exploding gradients and ensure stable training.

The choice of $\beta = 1 \times 10^{-4}$ for both robust methods represents a balance between model performance and robustness. Too large a value would overly emphasize worst-case scenarios, while too small a value would not provide sufficient robustness against uncertainty. Through empirical testing, this value demonstrated the best trade-off between trading performance and resilience to market fluctuations.

Advantage normalization is applied prior to the policy update to stabilize training and improve convergence. The PPO clip parameter $\epsilon = 0.2$ prevents excessively large policy updates, maintaining proximity to the previous policy while allowing sufficient exploration.

**Discretization of Execution Prices**    To apply the robust RL framework, we apply the discretization to the execution price. We introduce two additional hyper-parameter to control the discretization level: $2N + 1$ represents the number of total discretization in the execution prices and $\delta$ represents the strength of each level. Given the execution price $p$, we have $2N + 1$ potential execution prices $[p - N\epsilon, \ldots, p - \epsilon, p, p + \epsilon, \ldots, p + N\epsilon]$ in total. This discretization approach allows us to directly apply the closed-form solution to solve the optimal $u^*$. The uncertainty dimension and the epsilon in the robustness parameters are used to determine the discretization level of execution prices. For example, when the discretization level is given by 3, we have three potential execution prices $[p + \epsilon, p, p - \epsilon]$, which also corresponds to the uncertainty dimension. In the nominal transition kernel, we assume the distribution over this discretized set as $[0.25, 0.5, 0.25]$, which provides sufficient flexibility for us to choose the uncertainty $u$.

**Uncertainty Restriction to Execution Prices**    Instead of perturbing the whole transition probability, we restrict the perturbation on the execution price. The formulation of restriction is given as follows: we denote the state $s_t$ as two components $(p_t, f_t)$, and we hope perturb the transition on $p_t$ only. We re-write the transition probability using the conditional probability

$$\begin{aligned}
\mathbb{P}(s_{t+1} \mid s_t, a_t) &= \mathbb{P}(p_{t+1}, f_{t+1} \mid s_t, a_t) \\
&= \mathbb{P}(p_{t+1} \mid s_t, a_t)\mathbb{P}(f_{t+1} \mid p_{t+1}, s_t, a_t).
\end{aligned}$$

Then we set the uncertainty $u$ as the perturbation on the kernel $\mathbb{P}(p_{t+1} \mid s_t, a_t)$ instead of the whole transition kernel $\mathbb{P}(s_{t+1} \mid s_t, a_t)$. Given that the transition of the execution price $p_{t+1}$ is discretized, the transition probability $\mathbb{P}(p_{t+1} \mid s_t, a_t)$ is a discrete distribution. For example, when taking the uncertainty dimension as 3, we in fact obtain: $\mathbb{P}(p_{t+1} = p + \epsilon \mid s_t, a_t) = 0.25 + u^1$,

$\mathbb{P}(p_{t+1} = p \mid s_t, a_t) = 0.25 + u^2$, and $\mathbb{P}(p_{t+1} = p - \epsilon \mid s_t, a_t) = 0.25 + u^3$. By default, in the P1N2-type robust RL, we choose the shift parameter $u_1 = [u^1, u^2, u^3] = [0.1 - 1/3, -1/3, -1/3]$ if the action is buying a stock and $u_1 = [u^1, u^2, u^3] = [-1/3, -1/3, 0.1 - 1/3]$ if the action is selling a stock.

**Solving the Worst-Case Uncertainty**    After restricting the domain to the discrete execution price, we turn the original worst-case Bellman equation into the following form:

$$V^\pi(s) = r(s, a) + \gamma \min_u \sum_{s'} \sum_a V^\pi(s')\pi(a|s)\mathbb{P}_u(s'|s, a)$$

$$= r(s, a) + \gamma \min_u \sum_{s'} \sum_a V^\pi(s')\pi(a|s)\mathbb{P}_u(p' \mid s, a)\mathbb{P}(f' \mid p', s, a)$$

$$= r(s, a) + \gamma \min_u \sum_{s'} \sum_a V^\pi(s')\pi(a|s)\left[\mathbb{P}_0(p' \mid s, a) + u_{s,a}\right]\mathbb{P}(f' \mid p', s, a)$$

$$= r(s, a) + \gamma \sum_{s'} \sum_a V^\pi(s')\pi(a|s)\mathbb{P}_0(s'|s, a) + \gamma \min_u \sum_{s'} \sum_a u_{s,a} V^\pi(s')\pi(a|s)\mathbb{P}(f' \mid p', s, a).$$

Moreover, we use $\mathbb{P}(f' \mid p', s, a)$ as a deterministic transition. It is common when training in the historical data, as (1) the observed feature $f'$ comes from the existing dataset and the action $a$ does not change its value (in our implementation, the action $a$ will only affect the execution price), and (2) many features are directly calculated based on the current state, action, and the execution price such as the remaining cash. As the result, we obtain

$$V^\pi(s) = r(s, a) + \gamma \sum_{s'} \sum_a V^\pi(s')\pi(a|s)\mathbb{P}_0(s'|s, a) + \gamma \min_u \sum_{p',s'=(p',f')} u(p')V^\pi(s')$$

$$= r(s, a) + \gamma \sum_{s'} \sum_a V^\pi(s')\pi(a|s)\mathbb{P}_0(s'|s, a) + \gamma \min_u u^\top V^\pi.$$

Here we still write $u^\top V^\pi$ for convenience, while the value function $V^\pi$ represents the $2N + 1$ dimensional vector with value taken on $s' = (p', f')$, where $p'$ takes $2N + 1$ values and $f'$ is deterministically determined by $(p', s, a)$. After making this modification, we can solve $\min_u u^\top V^\pi$ using Theorem 3.4 and Theorem 3.5, or existing $\ell_p$-norm formula [39, 40].

