# OpenReview forum: "Robust Reinforcement Learning in Finance:  Modeling Market Impact with Elliptic Uncertainty Sets"
_NeurIPS.cc/2025/Conference — NeurIPS 2025 poster_

### Official Review · Reviewer_SFEg · 2025-06-07

**Clarity:** 2
**Significance:** 3
**Originality:** 3
**Rating:** 4
**Confidence:** 3

**Summary:**

The simulator built with historical data has a huge limitation in that the actions of the agent cannot influence the stock price. To address this issue, this paper has proposed a novel approach that model the market impact with elliptic uncertainty sets. The proposed approach has solid theoretical guarantees, and the empirical results have shown the effectiveness of the proposed approach.

**Questions:**

The authors are encouraged to clarify the theoretical analysis.

**Ethical Concerns:**

["NO or VERY MINOR ethics concerns only"]

**Limitations:**

Yes

**Quality:**

3

**Strengths And Weaknesses:**

This paper deals with an important problem, the market impact issue.

The proposed approach has a solid theoretical grounding. The authors have derived closed-form solutions under the proposed uncertainty sets. Beyond that, the authors have presented explicit solutions under certain conditions.

The experiments are conducted on real-world financial data, and the experiment results show improvement over existing RL baselines.

The theoretical analysis is a little hard to follow, and it is unknown whether the uncertainty in the theorem is consistent with that often occurs in the real market, i.e., are the assumptions in the theorem realistic?

---

> ### Author Rebuttal · Authors · 2025-07-29
>
> We sincerely appreciate the reviewer’s positive evaluation and constructive feedback. To improve the clarity and readability of the theoretical analysis, we will provide more detailed explanations at each step. Below, we address the remaining questions raised by the reviewer:
>
> > **Question:** The theoretical analysis is a little hard to follow, and it is unknown whether the uncertainty in the theorem is consistent with that often occurs in the real market, i.e., are the assumptions in the theorem realistic?
>
> **Response:** We thank the reviewer for this insightful question. Rather than referring to it as an "assumption", we prefer to name it as a "model"; that is, a model for describing the market impact. Obviously, all models are inaccurate on describing the real market. Therefore, we interpret the question as: *Does the theoretical model proposed in our work offer a more realistic representation of market impact compared to existing models?*
>
> We answer this question by splitting it into the following three parts:
>
> * *How to model the market impact with uncertainty sets?*
>
>     When we hope buy an asset with the price $p$, the actual execution price $p_e$ is typically different from $p$ due to the market impact. Instead of trainning a neural network to predict the execution price, the uncertainty set is used to model this difference by giving a range of it. Specifically, in traditional robust RL, we set $\beta=|p_e -p|$ and consider the ball $u\in [-\beta, \beta]$. It means the true execution price $p_e$ would belongs to the set $[p - \beta, p +\beta]$.  However, we observe that market impact tends to be directional. For instance, buying usually drives the price up, while selling pushes it down. As a result, it may be more realistic to model the execution price as $[p, p + \beta]$ when buying and $[p - \beta, p]$ when selling. Our elliptical uncertainty set captures this asymmetry while a ball can never make it.
>
> * *Is this approach more realistic compared to existing models?*
>
>     Yes. By incorporating directionality, our model better reflects typical market behaviors: When buying a stock, the actual execution price is usually higher than the current price; therefore, we can filter out some less realistic moves in our uncertainty set to make it more realistic. Moreover, the elliptical uncertainty set generalizes the ball-shaped uncertainty set, making it capable of modeling both symmetric and asymmetric impact patterns. This flexibility enhances its applicability to a wider range of market conditions.
>
> * *Which part is still unrealistic?*
>
>     We acknowledge that our model still simplifies certain aspects of market behavior. Two key limitations include: (i) The extreme market move. Even large trades may have negligible impact in highly liquid markets with abundant counter-orders.  In highly liquid markets with abundant counter-orders, even large trades may have negligible impact. For instance, during widespread selling pressure, a large buy order may still execute at a decreasing execution price. (ii) The discretization. We discretize execution prices using a finite grid, which introduces approximation error compared to continuous-valued prices.
>
>     Despite these limitations, we still believe the elliptic uncertainty contributes meaningfully to bridging the gap between simulated environments and real-world financial markets by addressing the intrinsic disadvantage of classical ball-shaped uncertainty set.
>
> We appreciate the reviewer again for the positive evaluation. We hope our explanation resolves your concern.

---

### Official Review · Reviewer_E9d3 · 2025-06-25

**Clarity:** 4
**Significance:** 2
**Originality:** 3
**Rating:** 4
**Confidence:** 3

**Summary:**

This paper tackles the market impact challenge in financial RL by introducing elliptic uncertainty sets that capture directional price shifts.
Contributions include
1) Developing a novel class of elliptic uncertainty sets better to capture the empirically observed directional nature of market impact.
2) Deriving implicit and explicit solutions for robust policy evaluation.
3) Evaluating on real-world financial data, results show that the new robust TD learning algorithm outperforms baselines in Sharpe ratio and remains robust under high volumes, offering a more effective RL framework for financial markets.

**Questions:**

1) In Example 3.1, the paper states that symmetric uncertainty sets fail to capture directional uncertainty. How do the proposed elliptic uncertainty sets capture the directional nature of environmental shifts in financial markets? It would be helpful to have intuitive examples or analyses similar to those for symmetric sets to better demonstrate the superiority of elliptic uncertainty sets.
2) The paper proposes an ellipse with N foci based on the classical ellipse. How is N determined in real financial market experiments, and what value of N achieves the best performance in practical applications?

**Ethical Concerns:**

["NO or VERY MINOR ethics concerns only"]

**Final Justification:**

I appreciate the authors for providing a comprehensive response, and I support the acceptance of this paper.

**Limitations:**

yes

**Paper Formatting Concerns:**

None.

**Quality:**

3

**Strengths And Weaknesses:**

Pros:
1) The paper investigates financial market characteristics and identifies market impact as a key challenge in robust RL, proposing elliptic uncertainty sets to address the limitations of traditional symmetric sets, thus focusing on a critical financial issue with innovative solutions.
2) The paper offers a thorough analysis of common uncertainty sets in the literature, complemented by comprehensive theoretical derivations of both implicit and explicit solutions for elliptic uncertainty sets.
3) The paper is well-structured and concisely written, ensuring clarity in organizing backgrounds and ideas and presenting technical content and proving.
Cons:
1) The paper lacks detailed derivations and development of robust RL-related methods, such as the robust Bellman operator and robust TD learning algorithm, which might be better if presented in the background.
2) The experiments use relatively few baselines, insufficient to fully demonstrate the method's superiority, and the analysis of results is overly simplistic.
3) The theoretical analysis relies heavily on finite state and action spaces, which may not hold in real-world financial markets with continuous dynamics, potentially limiting generalizability.
4) Experiments focus on historical data and standard market conditions, lacking validation under extreme scenarios, which limits real-world risk assessment.

---

> ### Author Rebuttal · Authors · 2025-07-29
>
> We deeply appreciate the reviewer’s positive evaluation and constructive feedback. Below, we provide our point-by-point responses to each of the concerns raised.
>
> ## Weaknesses
>
> > **W1:** The paper lacks detailed derivations and development of robust RL-related methods, such as the robust Bellman operator and robust TD learning algorithm, which might be better if presented in the background.
>
> **Response:** We appreciate the reviewer’s helpful suggestion. In the revised version, we will include additional background on robust RL.
>
> > **W2:** The experiments use relatively few baselines, insufficient to fully demonstrate the method's superiority, and the analysis of results is overly simplistic.
>
> **Response:** We appreciate the reviewer's suggestions. We would like to clarify that our experiments are conducted to validate the effectiveness of reducing the market impact instead of proposing a new RL trading strategy; therefore, adding more baselines may not be helpful in validating this statement.
>
> > **W3:** The theoretical analysis relies heavily on finite state and action spaces, which may not hold in real-world financial markets with continuous dynamics, potentially limiting generalizability.
>
> **Response:** We appreciate the reviewer’s insightful comment. We agree that the discretization may not fully capture the continuous dynamics of real-world financial markets. Bridging this gap between discrete theoretical models and continuous environments is an important and active area of research in modern RL, and we consider it a valuable direction for future work.
>
> > **W4:** Experiments focus on historical data and standard market conditions, lacking validation under extreme scenarios, which limits real-world risk assessment.
>
> **Response:** We appreciate the reviewer’s suggestion. We agree that evaluating performance under extreme scenarios is important for comprehensive risk assessment. However, such scenarios often exhibit significant distributional shift from the training data, making model performance in these regimes less representative or informative. Therefore, we didn't include experiments in this setting.
>
> ## Questions
>
> > **Q1:** In Example 3.1, the paper states that symmetric uncertainty sets fail to capture directional uncertainty. How do the proposed elliptic uncertainty sets capture the directional nature of environmental shifts in financial markets? It would be helpful to have intuitive examples or analyses similar to those for symmetric sets to better demonstrate the superiority of elliptic uncertainty sets.
>
> **Response:** We thank the reviewer for the thoughtful question and helpful suggestion. Below, we clarify how elliptic uncertainty sets capture directional uncertainty in financial markets, using the following simplified example:
>
> * Assume we are **buying** a stock at the price $100$. The execution price would be $\\\{99.8, 99.9, 100.0, 100.1, 100.2 \\\}$, which corresponds to the uncertainty set $\\\{-0.2, -0.1, 0.0, 0.1, 0.2\\\}$.  We recall the symmetric uncertainty set $\\\{u: |u|\leq 0.2 \\\}$: if it needs to include $0.2$ in this set, we must also include $-0.2$ by the symmetry.
> * However, in our case, we may hope model the execution price as $\\\{100.0, 100.1, 100.2 \\\}$ instead of $\\\{99.8, 99.9, 100.0, 100.1, 100.2 \\\}$, as when buying a stock, it is more likely to push the price up, leading to the non-symmetric behavior. In this case, we need to use the elliptic uncertainty set $\\\{u: |u|+|u-0.2|\leq 0.2 \\\}$. We can easily verify that both $-0.2$ and $-0.1$ are not in this set.
>
> More generally, we treat execution prices as random variables and define uncertainty sets over their probability distributions. The elliptic set incorporates a directional prior by filtering out the unlikely execution prices such as $-0.2$, introducing prior knowledge into the RL agent; such knowledge would be difficult to learn from historical data alone, as the historical data does not contain information about the agent's own market impact. We will revise the paper to include this intuitive example and clarify the modeling advantage of elliptic uncertainty sets.
>
> > **Q2:** The paper proposes an ellipse with N foci based on the classical ellipse. How is N determined in real financial market experiments, and what value of N achieves the best performance in practical applications?
>
> **Response:**  In our experiments, we fix the number of foci as $2$ and do not tune the number of foci; specifically, we fix one focus point at \\$0, representing the unperturbed case, and the other at \\$$\pm$0.1, representing a scenario where the execution price is impacted by \\$0.1 (as the $(s,a)$-uncertainty set is action-dependent, we use different foci for the buying and the selling action). We found that this two-point elliptical uncertainty set is sufficient to capture the directional nature of price impact in our setting, and using a larger $N$ may result in difficulty of solving the worst-case uncertainty (this is also the reason why we further derive the explicit closed-form solution in Theorem 3.4); therefore, we did not consider configurations with more than two foci.

---

> > ### Author Response · Authors · 2025-08-07
> >
> > Dear Reviewer E9d3,
> >
> > Thank you for taking the time to provide a thoughtful and positive evaluation of our paper. As the author-reviewer discussion period is ending soon, we would just like to kindly follow up to see if our rebuttal has addressed the concerns you raised in your review. We're happy to follow up if you have any further questions.
> >
> > Sincerely,
> > Authors

---

### Official Review · Reviewer_aH4Q · 2025-07-03

**Clarity:** 4
**Significance:** 3
**Originality:** 3
**Rating:** 4
**Confidence:** 2

**Summary:**

This work analyzes the limitations of traditional symmetric uncertainty sets and proposes a novel elliptic uncertainty set that better captures real-world market fluctuations. Based on this new formulation, the paper studies its theoretical properties and derives closed-form solutions under different norms. Furthermore, it introduces a TD method to evaluate policies within this uncertainty framework. Finally, simulation results are provided to validate the proposed algorithm.

**Questions:**

My main concerns focus on the following two points:

1. When determining the worst-case transitions under an $l_p$ norm uncertainty set, as proposed in the paper, it is unavoidable that the resulting worst-case transition may not be a valid probability distribution. In particular, the closed-form solution can include negative probability values, which are not meaningful. This issue could significantly degrade the model's performance in practice. In financial applications—where interpretability and robustness are critical—such behavior is particularly problematic. Could the authors elaborate on the practical reasonableness of such results? How might this issue be mitigated or avoided?

2. Compared with uncertainty sets defined via $f$-divergences, the elliptic uncertainty set does not guarantee that the resulting distributions are absolutely continuous with respect to the nominal transition distribution. This raises a concern: could it be the case that, under certain conditions, the worst-case transition does not exist or is not meaningful in real-world applications? Are there specific and common conditions under which this lack of absolute continuity is acceptable or justified?

**Ethical Concerns:**

["NO or VERY MINOR ethics concerns only"]

**Final Justification:**

According to the review comments and rebuttal, I keep my borderline accept rating.

**Limitations:**

Yes, the authors adequately addressed the limitations and potential negative societal impact of their work.

**Paper Formatting Concerns:**

Nan

**Quality:**

3

**Strengths And Weaknesses:**

***Strengths***
1. The paper is clearly written. It highlights the limitations of traditional uncertainty sets and provides illustrative examples, which are especially helpful for readers who may not be familiar with the financial context.

2. The paper proposes a novel elliptic uncertainty set and offers solid and significant analytical results under this formulation.

***Weaknesses***
1. While the elliptic uncertainty set can mitigate transition mismatches observed in practice, it may introduce new types of mismatch—particularly when applying the closed-form solution.

2. The formulation in the reinforcement learning (RL) section could be more rigorous. For instance, in Line 92, the proper term is "$s,a$-rectangular uncertainty set." In Line 108, the correct terminology is "visitation distribution" to distinguish it from the stationary distribution. Additionally, a formal mathematical definition of this concept is missing. I recommend that the authors refer to the latest edition of the textbook [1] and the recent paper [2] to improve the precision of the presentation.

[1] Sutton, Richard S., and Andrew G. Barto. Reinforcement learning: An introduction. Vol. 1. No. 1. Cambridge: MIT press, 1998.

[2] Agarwal, Alekh, et al. "On the theory of policy gradient methods: Optimality, approximation, and distribution shift." Journal of Machine Learning Research 22.98 (2021): 1-76.

---

> ### Author Rebuttal · Authors · 2025-07-29
>
> We deeply appreciate the reviewer’s positive evaluation and constructive feedback. We have many thanks to reviewer's helpful suggestions on improving  rigor and clarity in the formulation; we will replace the imprecise phrasing with the correct terms and formal mathematical definitions.  We will also consult the recommended references to align our notational conventions and definitions with current standards in the literature.
>
> Below, we provide our point-by-point responses to each of the concerns raised.
>
> > **Q1:** When determining the worst-case transitions under an $\ell\_p$ norm uncertainty set, as proposed in the paper, it is unavoidable that the resulting worst-case transition may not be a valid probability distribution. In particular, the closed-form solution can include negative probability values, which are not meaningful. This issue could significantly degrade the model's performance in practice. In financial applications—where interpretability and robustness are critical—such behavior is particularly problematic. Could the authors elaborate on the practical reasonableness of such results? How might this issue be mitigated or avoided?
>
> **Response:** We appreciate the reviewer's careful reading and insightful question. Simply speaking, we will need to make an assumption to ensure each $u$ in the uncertainty set is well-defined; that is,  their induced transition probability $\mathbb{P}=\mathbb{P}_0+u$ presents no negative values).  This assumption is stated on Lines 100–101 and is consistent with prior theoretical work on $\ell_p$-norm uncertainty sets [kumar2023a,kumar2023b].
>
> * [kumar2023a] Kumar, Navdeep, et al. "An efficient solution to s-rectangular robust markov decision processes." *arXiv preprint arXiv:2301.13642* (2023).
> * [kumar2023b] Kumar, Navdeep, et al. "Policy gradient for rectangular robust markov decision processes." *Advances in Neural Information Processing Systems* 36 (2023): 59477-59501.
>
> We agree with the reviewer that our presentation on this assumption in our paper could be unclear.  In the revised version, we will explicitly label it as a formal assumption and clarify that the uncertainty set should be restricted accordingly to ensure the all uncertainties are well-defined.
>
> > **Q2:** Compared with uncertainty sets defined via $f$-divergences, the elliptic uncertainty set does not guarantee that the resulting distributions are absolutely continuous with respect to the nominal transition distribution. This raises a concern: could it be the case that, under certain conditions, the worst-case transition does not exist or is not meaningful in real-world applications? Are there specific and common conditions under which this lack of absolute continuity is acceptable or justified?
>
> **Response:** We thank the reviewer for raising this important point. Indeed, when using uncertainty sets that are not defined via $f$-divergences, the resulting worst-case transition distributions are not guaranteed to be absolutely continuous with respect to the nominal distribution. This can lead to pathological cases where no positive $\beta$ yields a valid (non-negative) transition probability. For instance, if the nominal transition is a delta distribution (i.e., assigns probability 1 to a single next state), then any perturbation under an $\ell_p$-norm may fail to produce a valid probability vector, as noted by the reviewer in the previous question.
>
> This limitation is intrinsic to the $\ell_p$-norm uncertainty set and has not been fully addressed in prior work (e.g., [kumar2023a, kumar2023b] mentioned above). As a result, we follow the convention in the literature and assume the nominal transition distribution is fully supported over the state space, which also ensures that the uncertainty set contains at least one valid perturbation, and hence the worst-case transition remains well-defined. In practice, this assumption is often satisfied. For example, in financial modeling, stock prices are commonly modeled using geometric Brownian motion (i.e., the Black–Scholes model), which yields a strictly positive density for the normalized price and implies full support over the state space. Therefore, the issue of non-existence due to lack of absolute continuity may not pose a practical limitation in financial domains.

---

> ### Comment · Reviewer_aH4Q · 2025-08-06
>
> Thank you for the authors' response. I acknowledge that this assumption is clearly stated in the paper. However, my concern lies in its reasonableness, particularly given that the study is situated in a financial setting. Specifically, if the minimum support (i.e., the smallest non-zero probability) is small, wouldn't the uncertainty level also need to be correspondingly small in order for the assumption to hold?
>
> In distributionally robust reinforcement learning (DRRL), such assumptions can sometimes be avoided by constraining the uncertainty set to lie within the true distribution set. However, this often results in a closed-form expression for the optimal distribution that is complex and intractable. Under these circumstances, analyzing temporal-difference (TD) learning becomes significantly more challenging. I understand that this motivates the assumption, but it does limit the practical applicability of the approach.
>
> Given these considerations, I will maintain my borderline rating.

---

> > ### Author Response · Authors · 2025-08-06
> >
> > We sincerely thank the reviewer for their time and thoughtful evaluation of our paper. We are encouraged by your continued positive evaluation of our submission.
> >
> > We fully agree that the full support assumption imposes a restriction on the uncertainty level, which is indeed an inherent limitation of the $\ell_p$-norm uncertainty sets in robust RL. As the reviewer rightly points out, this issue can sometimes be mitigated within the DRRL framework, albeit at the cost of "significantly more challenging" TD-learning. We will incorporate this valuable insight into our revision, including a discussion of related literature that explores DRRL approaches. We hope it will help motivate further research on addressing this trade-off.
> >
> > Once again, we appreciate the reviewer’s feedback in helping us improve our manuscript.

---

### Official Review · Reviewer_73NP · 2025-07-04

**Clarity:** 4
**Significance:** 2
**Originality:** 2
**Rating:** 4
**Confidence:** 4

**Summary:**

This paper proposes a robust RL approach particularly motivated by the lack of consideration about market impact in finance applications. The authors generalize the uncertainty set of missspecficiation the transition kernel from a unimodal definition to "elliptic" by introducing multiple foci and respective sizes. Based on this new uncertainty set, this paper then develop theoretical analysis of the Bellman update LP problem, and develop a TD learning algorithm. Numerical experiments on minute-level single-asset strategy and large-volume portfolio rebalancing are conducted.

**Questions:**

- Can the authors comment more on how the foci and set size can be decided? Ideally, can it be selected in a data-driven fashion, yet, not computationally intensively as cross validation?
- Can the authors provide a more theoretical characterization of how the elliptic uncertainty sets overcome the limitations of simple uncertainty sets other than the simple 5-dim examples?
- Why specifically focus on TD-learning algorithm? Any particular benefit?

**Ethical Concerns:**

["NO or VERY MINOR ethics concerns only"]

**Final Justification:**

I commend the authors' detailed and thoughtful response, and increased my score to reflect the improvement.

**Limitations:**

- The authors mention that their analysis only works with finite state and action space, which may have some discrepancy with real-world financial applications.

**Quality:**

3

**Strengths And Weaknesses:**

*Strength*
- The writing is very clear and polished. I like how the problems are well motivated and cleanly defined. Figure 2 is a nice explanation.
- The theorem presentation and theoretical analysis seem very rigorous with details.
- The numerical experiments are very detailed.

*Weakness*
- The overall methodology is quite straightforward and thus technical novelty is limited.
- The motivation of market impact, though interesting, is a bit niche.
- It is particularly unclear why the proposed robust optimization approach is a good solution to the proposed problem of market impact. Intuitively, the environment change due to market impact is potentially structured and directional, and it’s not immediately clear whether a worst-case robust approach, even one with an elliptic structure, captures that structure optimally. It would be helpful to better justify why robustness is a suitable solution and not overly conservative.

---

> ### Author Rebuttal · Authors · 2025-07-27
>
> We deeply appreciate the reviewer’s thoughtful and constructive feedback. Below, we provide our point-by-point responses to each of the concerns raised.
>
> ## Questions
>
> > **Q1:** Can the authors comment more on how the foci and set size can be decided? Ideally, can it be selected in a data-driven fashion, yet, not computationally intensively as cross validation?
>
> **Response:** In our experiments, the foci and set size are treated as hyperparameters. Specifically, we fix one focus at \\$0, representing the unperturbed case, and the other at \\$$\pm$0.1, representing a scenario where the execution price is impacted by \\$0.1 (as the $(s,a)$-uncertainty set is action-dependent, we use different foci for the buying and the selling action). For the set size, we select from a predefined set of candidate values $\\\{ 0.1, 0.01, 0.001, 0.0001, 0.00001 \\\}$ and choose the one that yields the best performance on the training period (May 9th, 2021 to May 9th, 2022).
>
> We acknowledge the reviewer's concern regarding the computational cost of hyperparameter tuning described above. To address this issue, we believe the selection process can be made more data-driven by leveraging established techniques from finance and deep learning. In particular, financial models [1] such as Kyle’s Lambda or the Square-Root Impact Model can be used to estimate the expected market impact, which can then inform the choice of foci. The corresponding uncertainty around this estimate can be derived from traditional statistical confidence intervals. Alternatively, modern deep learning models could be employed to learn these parameters directly from data, offering a potentially more scalable and adaptive solution. As the result, we believe this direction can naturally connect to many existing works in data-driven finance and market microstructure modeling.
>
> * [1] Hasbrouck, Joel. "Trading costs and returns for US equities: Estimating effective costs from daily data." *The Journal of Finance* 64.3 (2009): 1445-1477.
>
>
>
> > **Q2:** Can the authors provide a more theoretical characterization of how the elliptic uncertainty sets overcome the limitations of simple uncertainty sets other than the simple 5-dim examples?
>
> **Response:** From a theoretical perspective, we characterize the conservativeness of an uncertainty set using the robust value function. That is, if the robust value function is smaller, then the uncertainty set is more conservative.  In the ideal scenario, suppose the true market impact is precisely represented by a specific perturbation vector $u_0$. If we define  $\mathcal{U}=\\\{ u_0\\\}$ as the uncertainty set; then robust RL over this singleton set exactly recovers the MDP induced by the true market dynamics. In this case, optimizing over the robust MDP is equivalent to training directly on an environment that accurately simulates the market impact.
>
> However, in practice, we cannot know the exact value of $u_0$. We set $\beta = \\\|u_0\\\|+\epsilon$ as the size of the uncertainty set. Then both the ball $\mathcal{U}\_b=\\\{ u: \\\|u\\\| \leq \beta\\\}$ and the ellipse $\mathcal{U}\_e=\\\{u: \\\|u\\\|+\\\|u-u_0\\\| \leq \beta \\\}$ can contain the true value $u_0$, and moreover, we have $\mathcal{U}\_e \subset \mathcal{U}\_b$. This inclusion relation leads to the inequality
> $$\min\_{u \in \mathcal{U}\_b} V^\pi\_u\leq \min\_{u \in \mathcal{U}\_e} V^\pi\_u \leq V\_{u\_0}^\pi$$
> This suggests that the robust value function for the ball uncertainty set $\min\_{u \in \mathcal{U}\_b} V^\pi\_u$ is theoretically more conservative (more precisely, it is not less conservative) compared with the ellptic uncertainty set $ \min\_{u \in \mathcal{U}\_e} V^\pi\_u$.
>
>
>
> > **Q3:** Why specifically focus on TD-learning algorithm? Any particular benefit?
>
> **Response:** Typically, in traditional RL, there are two methods to estimate the value function: (i) *Monte-Carlo method*, also known as REINFORCE. A complete (or truncated) trajectory is sampled, and its discounted return is used as an unbiased but high‑variance estimate of the value. (ii) *TD-learning method (and its variants)*.  The agent bootstraps from a one‑step look‑ahead, repeatedly minimizing the TD‑error. This yields lower‑variance, more sample‑efficient updates.
>
> In robust RL, the TD-learning method  become almost unavoidable. Recall that our goal is to solve the robust value function which is defined in the form of $\min\_u V\_u^\pi$. The $\min\_u$ term make it challenging to directly apply the Monte-Carlo approach, as sampling from the worst-case transition requires to take an additional step to approximate it before sampling from it. As the result, the TD-learning method is much more preferable. After obtaining an estimate of the robust value function via TD learning, we can plug it into any standard policy‑gradient scheme (e.g. the PPO algorithm) using the generalized advantage estimate (GAE).
>
>
>
> ## Weaknesses
>
> We thank the reviewer again and would like to take this opportunity to clarify several points regarding the identified weaknesses.
>
> > **W1:** The overall methodology is quite straightforward and thus technical novelty is limited.
>
> **Response:** We thank the reviewer for recognizing the clarity of our methodology. We would like to emphasize that our work goes beyond a straightforward combination of robust RL and financial applications. Specifically, we introduce a novel perspective on modeling market impact. Prior studies have typically approached this by either (i) approximating the underlying MDP using fine-grained limit order book (LOB) data, or (ii) simulating market dynamics through large numbers of interacting agents. These methods aim to learn a direct mapping from observed data to execution prices. In contrast, our approach models market impact through uncertainty sets within the robust MDP framework, which provides a new perspective for understanding the market impact. Furthermore, we extend this framework by proposing an elliptical uncertainty set tailored to financial settings, addressing the drawback of traditional uncertainty set used in the RL literature when applied to the financial application. Lastly, we extend the closed-form solution of $\ell_p$-norm uncertainty set to our generalized ellipstic uncertainty set, which is also a new result even in the robust RL literatue.
>
>
>
> > **W2:** The motivation of market impact, though interesting, is a bit niche.
>
> **Response:** Instead of being niche, we tend to believe the topic is still emerging and growing in importance.
>
> * (i) On the finance side: With the field of modern quantitative trading is becoming increasingly crowded and competitive, understanding market impact has become a central concern in algorithmic trading and portfolio optimization.  By proposing a robust RL framework that better and easier models market impact, our work contributes to this growing body of research and provides tools that are under-explored but potentially applicable to real-world trading systems.
> * (ii) On the RL side: We are also interested in extending the application of the elliptical uncertainty set to other domains where RL has already played a central role, such as robotics and control. The geometric flexibility of the ellipse allows for modeling more directional or correlated uncertainties, which may benefit certain real-world systems with structured noise or anisotropic dynamics.
>
>
>
> > **W3:** It is particularly unclear why the proposed robust optimization approach is a good solution to the proposed problem of market impact. Intuitively, the environment change due to market impact is potentially structured and directional, and it’s not immediately clear whether a worst-case robust approach, even one with an elliptic structure, captures that structure optimally. It would be helpful to better justify why robustness is a suitable solution and not overly conservative.
>
> **Response:** We deeply appreciate the reviewer's insightful question. Capturing the optimal structure is indeed a challenging question and our approach cannot (and is not designed to) solve it. Instead, our method tells how to make it better when we cannot accurately obtain the execution price. Let's break down this idea in pieces:
>
> * The robust value function $V^\pi:=\min_{u\in\mathcal{U}} V_u^\pi$ models the worst-case discounted future return. If the uncertainty set $\mathcal{U}$ is larger, it is more conservative (as it gives smaller value). If the set $\mathcal{U}$ is smaller, it is less conservative.
> * The ideal case is that there exists the uncertainty $u_0$ exactly characterizing the MDP induced by the ground truth market impact and we directly use $\mathcal{U}=\\\{ u_0\\\}$ as the uncertainty set. In this case, training a RL agent on the robust MDP defined over $\mathcal{U}=\\\{ u_0\\\}$ is equivalent to train over the real LOB data.
> * The less ideal case is that even if the uncertainty set $\mathcal{U}$ contains some additional uncertainties; e.g. $\mathcal{U}=\\\{ u_0, u_1\\\}$, the minimum value $\min\_{u \in \mathcal{U}}$ is still achieved exactly at $u\_0$. In this case, the robust value function recovers to the previous ideal case. Otherwise, the robust value function gives a strictly smaller value than $V^\pi\_{u\_0}$, which makes it more conservative.
>
> Unfortunately, our approach cannot theoretically rule out this “otherwise” scenario. However, it is still guided by the above intuition: the smaller the uncertainty set, the less likely it is to include the conservative uncertainty. Our method contributes by trimming down the traditional uncertainty sets used in robust RL, thereby reducing the risk of being overly conservative.

---

### Public Comment · ~Shaocong_Ma1 · 2026-01-22
**Analysis and Fixes of Hallucinated Citations**

Dear all,

We sincerely apologize for the inclusion of hallucinated references in this paper. These miscitations are identified by the GPTZero team. The full report is available at:

> Nazar Shmatko, Alex Adam and Paul Esau. GPTZero finds 100 new hallucinations in NeurIPS 2025 accepted papers. January 2026. https://gptzero.me/news/neurips/

We found that these incorrect references are from our use of a large language model (LLM), to assist in generating bibliography entries from author-year citations, titles, or paraphrased descriptions. As a consequence, several references were hallucinated, leading to incorrect or non-existent BibTeX entries. Importantly, these references appeared only in the first sentence of the paper and were intended purely for high-level background context. Their removal does not affect the content, results, or conclusions of the paper; we are removing all identified citations and are submitting an updated version to arXiv (https://arxiv.org/abs/2510.19950). We will shortly reach out to the program chairs of NeurIPS'25 as well.

We deeply appreciate the GPTZero team for bringing this matter to our attention and we hope again sincerely apologize to the whole community.

### Affected Citations:

(Used for supporting the use of RL in market making)

* [23] Kyung Hyun Park, Hyeong Jin Kim, and Woo Chang Kim. Deep reinforcement learning for limit order book-based market making. Expert Systems with Applications, 169:114338, 2021.

* [24] Pierre Casgrain, Anirudh Kulkarni, and Nicholas Watters. Learning to trade with continuous action spaces: Application to market making. arXiv preprint arXiv:2303.08603, 2023.

(Used for supporting the use of RL in option hedging)

* [25] Petter N Kolm, Sebastian Krügel, and Sergiy V Zadorozhnyi. Reinforcement learning for optimal hedging. The Journal of Trading, 14(4):4-17, 2019.

* [27] H Cao, Y Wang, and Y Zhang. Risk-averse reinforcement learning for optimal option hedging. Journal of Computational Finance, 24(2):1-31, 2020.

* [28] W L Chan and R O Shelton. Can machine learning improve delta hedging? Journal of Derivatives, $9(1): 39-56,2001$.

* [29] Z Ning and Y K Kwok. Q-learning for option pricing and hedging with transaction costs. Applied Economics, 52(55):6033-6048, 2020.

---

### Note · Authors · 2025-08-15

Dear Reviewers and AC/SAC,

Thank you for the thoughtful reviews and for engaging during the discussion period. Below is a concise summary of the post-discussion outcome and the edits we plan to incorporate.

### Post-discussion outcome:

After rebuttal, **all reviews are unanimously on the positive side**.

### Reviewer-specific recap from the discussion:

*  *Reviewer 73NP*:  Initially noted that our presentation was unclear on why "the proposed robust optimization approach is a good solution to the proposed problem". After our rebuttal, the reviewer stated that our responses **"well addressed"** the main concern and **raised their score**.

* *Reviewer aH4Q*: Expressed concerns about "mitigating transition mismatches" and encouraged the use of more standard RL notation. We will adopt the suggested, more rigorous notation in our revision. After our rebuttal, the reviewer maintained a **positive evaluation** in support of our work.

* *Reviewer E9d3*:  Asked why symmetric uncertainty sets fail to capture market impact. We provided an illustrative example to clarify this, and the reviewer maintained a **positive score**.

* *Reviewer SFEg*: Encouraged us to "clarify the theoretical analysis". We addressed this in our rebuttal and will incorporate these clarifications into the revised manuscript, and the reviewer kept a **positive score** post-discussion, showing support for our contribution.

---

We greatly appreciate the constructive feedback and believe these planned edits will meaningfully strengthen the paper.

Sincerely,

The Authors

---

### Decision · Program_Chairs · 2025-09-17

**Decision:**

Accept (poster)

**Comment:**

Motivated by the mismatch in financial RL between training on historical data and deployment in live markets (where trading actions cause market impact), the authors introduce a class of elliptic uncertainty sets that better capture the directional nature of market impact, deriving efficient closed-form solutions for robust policy evaluation. Experiments on single- and multi-asset trading tasks show improved Sharpe ratios and robustness under higher trade volumes, offering a scalable approach for financial RL.

The paper is very well-written and the results are solid. There were initial concerns centered around the motivation, technical novelty, and the limitation on tabular MDPs, but the reviewers remained (or became more) positive about the paper after the discussion period. Overall, I recommend a borderline accept.